# Comparison of Antibiotic Resistance Mechanisms in Antibiotic-Producing and Pathogenic Bacteria

**DOI:** 10.3390/molecules24193430

**Published:** 2019-09-21

**Authors:** Hiroshi Ogawara

**Affiliations:** 1HO Bio Institute, 33-9, Yushima-2, Bunkyo-ku, Tokyo 113-0034, Japan; hogawara@sc5.so-net.ne.jp; Tel.: +81-3-3832-3474; 2Department of Biochemistry, Meiji Pharmaceutical University, 522-1, Noshio-2, Kiyose, Tokyo 204-8588, Japan

**Keywords:** antibiotic resistance, self-resistance, bacterium, antibiotic producer, pathogenic bacteria

## Abstract

Antibiotic resistance poses a tremendous threat to human health. To overcome this problem, it is essential to know the mechanism of antibiotic resistance in antibiotic-producing and pathogenic bacteria. This paper deals with this problem from four points of view. First, the antibiotic resistance genes in producers are discussed related to their biosynthesis. Most resistance genes are present within the biosynthetic gene clusters, but some genes such as paromomycin acetyltransferases are located far outside the gene cluster. Second, when the antibiotic resistance genes in pathogens are compared with those in the producers, resistance mechanisms have dependency on antibiotic classes, and, in addition, new types of resistance mechanisms such as Eis aminoglycoside acetyltransferase and self-sacrifice proteins in enediyne antibiotics emerge in pathogens. Third, the relationships of the resistance genes between producers and pathogens are reevaluated at their amino acid sequence as well as nucleotide sequence levels. Pathogenic bacteria possess other resistance mechanisms than those in antibiotic producers. In addition, resistance mechanisms are little different between early stage of antibiotic use and the present time, e.g., β-lactam resistance in *Staphylococcus aureus*. Lastly, guanine + cytosine (GC) barrier in gene transfer to pathogenic bacteria is considered. Now, the resistance genes constitute resistome composed of complicated mixture from divergent environments.

## 1. Introduction

The introduction of antibiotics once reduced human morbidity and mortality caused by infectious diseases dramatically. For example, human morbidity and mortality by tuberculosis were greatly reduced after the introduction of streptomycin and kanamycin. However, the emergence of multidrug resistant pathogenic bacteria reverted the situation drastically once again. Moreover, the situation of multidrug resistance is getting worse and worse. WHO calls attention to the infections especially by *Klebsiella pneumoniae*, *Mycobacterium tuberculosis*, and *Neisseria gonorrhoeae*, and blood poisoning and foodborne diseases, where these infections are becoming harder and sometimes impossible to treat [1]. Antibiotic resistance is mainly due to efflux of antibiotics by transporters, prevention of interaction of antibiotics with target by mutation, modification and protection of target, and modification of antibiotics. These mechanisms result from the inherent structural or functional resistant characteristics, the acquired resistance by mutational change or horizontal gene transfer, and the adaptive antibiotic resistance [2,3,4]. The fact that even before the rediscovery of penicillin resistant bacteria were reported suggests strongly that at least some resistance traits were intrinsically present in their genomes [5,6]. Together with the natural conception that antibiotic-producers should possess the self-resistance mechanisms for the prevention of their suicide, it was hypothesized that the genes for antibiotic modifying enzymes evolved in the antibiotic-producing bacteria and were transferred to pathogenic bacteria through transformation, transduction or conjugation [7,8,9]. This hypothesis has been accepted without dispute until recently [10]. However, the long-time misuse and overuse of antibiotics have resulted in the widespread dissemination of antibiotics as well as antibiotic resistance genes all over the environment, not only in sewage and wastewater treatment plants, hospital effluents, aquaculture, agricultural and slaughterhouse waste, but also in surface waters, soils, and so on. Consequently, multidrug resistance traits have migrated reciprocally among the various bacteria residing in these environments [11,12,13,14]. This review paper summarizes first the antibiotic resistance genes in producing bacteria from the point of view of antibiotic biosynthesis. Then the resistance genes in pathogenic bacteria are compared with those in the producers. Lastly, the relationships of the resistance genes between producing bacteria and pathogenic bacteria are reevaluated again at their amino acid sequence as well as nucleotide sequence levels.

## 2. Protein Synthesis Inhibitors

### 2.1. Aminoglycosides

On the basis of the mechanism of action, antibiotics can be classified into six categories: protein synthesis inhibitors, cell wall synthesis inhibitors, DNA synthesis inhibitors, DNA intercalators, RNA synthesis inhibitors, and others. Protein synthesis is one of the major targets for antibiotics in the cell, and can be divided into four distinct phases: initiation, elongation, termination and recycling [15,16,17]. Aminoglycoside antibiotics belong to the protein synthesis inhibitors, and are grouped into 4,6-disubstitued 2-deoxystreptamine (DOS)-containing (kanamycin, tobramycin, gentamicin, and others), 4,5-disubstitued DOS-containing (neomycin, paromomycin, lividomycin, and others), 4-monosubstitued DOS-containing (apramycin) aminoglycosides, and others (streptomycin, spectinomycin, hygromycin B, kasugamycin, and others). These aminoglycosides exert their antimicrobial action by binding primarily to helix 44 of 16S rRNA of the small subunit of bacterial ribosome in the decoding region (A-site) and secondarily to helix 69 of 23S rRNA of the large subunit, leading to the induction of translational misreading and inhibition of the translocation reaction [18,19,20]. In addition, streptomycin, one of the atypical aminoglycosides, interacts with the backbone phosphates and ribose hydroxyl groups. One characteristic of aminoglycoside antibiotics is that they are considered as a bactericidal class of antibiotics [18,21], in contrast to the fact that most antibiotics that target the ribosome such as macrolides, tetracyclines, and chloramphenicol are bacteriostatic [22,23,24]. The detailed mechanisms of action of aminoglycosides are depending on their sophisticated chemical structures [25,26,27,28].

These aminoglycoside antibiotics are biosynthesized by *Actinobacteria*. *Actinobacteria* are prokaryotes, so they must protect themselves against attacks by their own biosynthetic products. Appendix A shows antibiotic names, their chemical structures (Kegg numbers), antibiotic resistance-related strategies in producing bacteria/fungi, analyzed bacteria/fungi, and references and GenBank accession numbers (GB No.). Kanamycin was isolated from *Streptomyces kanamyceticus* [29] and is a 4,6-disubstitued DOS-containing aminoglycoside antibiotic, together with tobramycin, gentamicin and sisomicin. Dibekacin, amikacin, netilmicin, and isepamicin are their semisynthetic analogues. The kanamycin biosynthetic gene clusters were cloned and sequenced [30,31] (GB Nos. AJ582817, AB164642, and AB254080). They contain genes for aminoglycoside 6′-N-acetyltransferase (*kanM*) and 16S rRNA methyltransferase (*kmr*), indicating that they are involved in the self-resistance. In addition, there are several efflux (*kanO* and *kanN*) and ABC transporter protein genes (*kanS*, *kan R* and *kanQ*). Tobramycin is 3′-deoxykanamycin B. The biosynthetic gene clusters were cloned from *Streptoalloteichus tenebrarius* and *Streptoalloteichus hindustanus* [30,32,33] (GB Nos. AJ579650, and AJ810851). Two transporter genes (*tobT* and *tobU*) are present, but acetyltransferase and phosphotransferase genes [34] are absent within the clusters. A tobramycin producer, *S*. *tenebrarius*, was reported to produce rRNA methyltransferase (KgmB) [35]. These genes may be involved in the self-resistance. The gentamicin biosynthetic gene clusters were cloned from *Micromonospora inyoensis* [36,37] (GB Nos. AJ575934, AJ628149, AY524043]. Interestingly, four genes proposed to be involved in the self-resistance are present within the clusters, that is, the genes encoding two rRNA methyltransferases (*gtmF* and *gtmL*), one aminoglycoside phosphotransferase (*gtmJ*), and one transporter (*gtmK*). Sisomicin is structurally related to gentamicin but has a unique unsaturated diaminosugar. G-52 is 6-methylsisomicin and G-418 (geneticin) is structurally similar to gentamicin. The sisomicin biosynthetic gene cluster was cloned from *M*. *inyoensis* [38] (GB No. FJ160413). Similar to the gentamicin gene cluster, the sisomicin biosynthetic gene cluster contains the genes encoding two rRNA methyltransferases (*sis4* and *sis9*), one aminoglycoside phosphotransferase (*sis17*), and two but not one transporters (*sis26* and *sis27*). In addition, the arrangement of the genes resembles very much with each other. Analysis of cell-free extracts of G-52 and G-418 producers showed that they were devoid of modification enzymes specific for aminoglycosides. Instead, they contained aminoglycoside highly resistant ribosomes [39,40,41]. As for verdamicin and sagamicin, 4,5-disubstituted DOS-containing, *Micromonospora* species-producing aminoglycosides, no report has been published on their biosynthetic genes yet. Only methylated and phosphorylated metabolites were detected in the fermentation broth [42].

Neomycin (fradiomycin), paromomycin, and lividomycin belong to 4,5-disubstitued DOS-containing aminoglycoside antibiotics. Their clinical use is limited by their toxicity. The neomycin biosynthetic gene cluster was cloned from *S*. *fradiae* as a 37kb DNA fragment including 21 putative open reading frames [43,44,45] (GB No. AJ843080). They contain genes for AphA (GB No. CAF33306) and AacC8 (GB No, CAF33325) proposed to be involved in the self-resistance process. They are aminoglycoside 3′-phosphotransferase and aminoglycoside 3-acetyltransferase, respectively. In addition, genes for two putative ABC transporters (GB Nos. CAF33314 and CAF33315) were detected within the cluster. The search for the rRNA methyltransferase which may be involved in the self-resistance, using GrmA from *M*. *echinospora* (GB No. AAR98546), Kmr from *S*. *kanamyceticus* (GB No. CAE46946) and KamB (GB No. WP_063964000) from *S*. *tenebrasius* as probe proteins revealed that no similar protein is present in *S*. *fradiae* DSM40063, suggesting that no rRNA methyltransferase is detectable, although at least parts of the aminoglycoside biosynthetic genes are present in the genome. The paromomycin biosynthetic gene cluster was cloned from *Streptomyces rimosus* as a 48kb DNA fragment (GB No. AJ628955). The self-resistant aminoglycoside 3′-phosphotransferase (*parR*/*aphA*) and ABC transporter genes (*parT* and *parU*) are located within the cluster, and their amino acid sequences are highly homologous to those in neomycin biosynthetic gene cluster. Interestingly, two acetyltransferases (AAC(3)-VII and AAC(6′)-II, GB Nos. CAG44462 and CAG44463), which may be involved in the self-resistance, are present far outside the gene cluster [46] (GB No. AJ749845). It is intriguing to know how these enzymes are involved in the self-resistance. The lividomycin biosynthetic gene cluster was cloned from *Streptomyces lividus* as 40 kb DNA fragment (GB No. AJ748832). No resistance-related gene was reported in the producer except two ABC transporter genes (GB Nos. CAG38699 and CAG38700). This may be due to no detailed examination for the self-resistance gene in the strain. The ribostamycin biosynthetic gene cluster was published [47] (GB No. 744850). Similar to other 4,5-disubstituted aminoglycosides like neomycin, self-resistance-related genes, *rph*, *rbmI*, *rgmE* and *rbmF* were detected within the biosynthetic gene cluster. Butirosin is a 4,5-disubstituted DOS-containing aminoglycoside produced by *Bacillus circulans*. The chemical structure of butirosin is similar to ribostamycin except that a part of DOS is substituted by α-hydroxy-γ-aminobutyric acid, but the organization of their biosynthetic gene cluster is completely different between them [47,48,49] (GB No. AB097196). Moreover, although an aminoglycoside 3′-phosphotransferase activity was detected in a butirosin producer [50], such gene is not detectable in the biosynthetic gene cluster. Apramycin is a mono-substituted DOS-containing aminoglycoside. The biosynthetic gene cluster was cloned from *S*. *tenebrarius* (GB No. AJ629123). It contains genes for 16S rRNA methyltransferase (*kamB*) and two exporters (*aprV* and *aprW*), which may be involved in the self-resistance. The antimicrobial activities of these 4,6- and 4,5-disubstituted and monosubstituted DOS-containing aminoglycosides are compromised by methylation of G1405 or A1408 in helix 44 of 16S rRNA [51].

Streptomycin is the first antibiotic isolated from *Actinobacteria* and the first aminoglycoside antibiotic [52]. Different from the aminoglycosides described above, streptomycin binds to the four nucleotides of 16S rRNA (Nos. 13, 526, 915, and 1490) and lysine45 of protein S12, and thereby its antibacterial activity is not affected by the methylation of G1405 and A1408 [53]. The biosynthetic gene cluster was cloned as 90kb DNA fragment [54,55] (GB No. AJ862840). It contains three putative phosphotransferase genes (*strA*/*aphD*/SGR_5932, *strK*/SGR_5938, and *aphE*/SGR_249) and two ABC-type transporter genes (*strV*/SGR_5915 and *strW*/SGR_5916) located adjacent to the cluster. In addition, one putative aminoglycoside acetyltransferase (SGR_292) and four RNA methyl-transferases are detected (SGR_1654, SGR_1886, SGR_4020, and SGR_6774). Not all but at least some of these proteins are implicated in the self-resistance in the producer [56] (GB No. NC_010572). The biosynthetic gene cluster of an aminocyclitol aminoglycoside antibiotic spectinomycin was cloned from *S*. *spectabilis* and other *Streptomyces* species [57,58] (GB No. EU255259). An aminoglycoside phosphotransferase, SpcN and an RNA methyltransferase, SpcM, were proposed to be involved in the self-resistance [57,58]. Hygromycin B is an aminocyclitol antibiotic that binds to discrete sites on the 30S ribosomal subunit and inhibits protein synthesis [59]. The biosynthetic gene cluster was cloned from *S*. *hygroscopicus* [60,61] (GB No. AJ628642). The self-resistance-related proteins, HygA (phosphotransferase) and transporters (HygV and HygW), are detected within the cluster. Hygromycin A, structurally distinct from hygromycin B, is an antibiotic isolated from *S*. *hygroscopicus* and inhibits the peptidyltransferase reaction of protein synthesis. The biosynthetic gene cluster of 31.5kb DNA fragment was cloned [62]. Hyg21, a phosphotransferase, and Hyg19 and Hyg28, transporters, were proposed to be involved in the self-resistance [62,63]. Istamycin produced by *S*. *tenjimariensis* is an aminoglycoside antibiotic composed of two units. FmrT consisting of 211 amino acid residues was proposed to be the rRNA methyltransferase involved in the self-resistance [64]. Three transporter genes (*steF24.1*, *steF24.27c*, and *steO22.6*) were speculated within the cluster. Istamycin is also acetylated by kasugamycin-producing *S*. *kasugaensis* [65]. However, the relation of the acetylation to the self-resistance in the istamycin-producer has not been clarified. Kasugamycin is an aminoglycoside antibiotic mainly used for the prevention of the growth of a fungus causing rice blast disease, and specifically inhibits translation initiation [66]. Acetyltransferase activity was reported to be involved in the self-resistance in kasugamycin-producing *S*. *kasugaensis* [65]. In addition, the ABC transporter genes, *kasKLM*, are responsible for the self-resistance [67]. Fortimicin (astromicin) is an aminoglycoside antibiotic produced by *Micromonospora olivasterospora*. The biosynthetic gene cluster was cloned (GB No. AJ628421) and, *fmrO* encoding 16S rRNA methyltransferase (GB No. CAF31555) was assumed to play a role in the self-resistance [68]. Validamycin is a fungicidal aminoglycoside antibiotic produced by *S*. *hygroscopicus* var. *limoneus*. It is used as an inhibitor of trehalase. The biosynthetic gene cluster was cloned [69] (GB No. DQ223652). A putative transporter (VldJ) is present within the cluster. A derivative of aminoglycoside, acarbose was isolated from *Streptomyces diastaticus* as an amylase inhibitor [70]. It is now widely used for the treatment of patients with type 2 diabetes mellitus [71]. However, nothing has been reported on its antimicrobial activity. The biosynthetic gene cluster was cloned [72] (GB No. AM409314). Acarbose kinase, GacK, and three transporters, GacX, GacY, and Gac W, are present within the cluster. GacK was reported to be implicated in the intracellular inactivation of acarbose [72]. Acarbose is conceived to have a dual role for the producer, that is, that it inhibits α-glucosidic enzymes of competitors and functions as a carbophor for the uptake of glucose or starch molecules in the producer. Although the streptothricin class antibiotics, streptothricin and nourseothricin, show a broad antimicrobial activity, their characteristic delayed toxicity interrupts their clinical application. The cloned biosynthetic gene cluster comprises genes for an acetyltransferase (*orfE*) and two transporters (*orfW* and *orfX*). These proteins are supposed to be involved in the self-resistance [73,74,75] (GB Nos. AB684619 and AB684620). Nourseothricin acetyltransferase gene (*natI*), which is involved in the self-resistance, was cloned [76,77].

The drug resistance in the antibiotic-producing bacteria is limited, so to speak, to their own territories. In contrast, that in pathogenic bacteria not only affects their own existence, but also affords threatening effects on human and livestock. The resistance mechanism of pathogenic bacteria to aminoglycoside antibiotics includes aminoglycoside-modifying enzymes, the mutation and the modification of the ribosomal target, and efflux pumps [78,79]. However, the most widely disseminated means of resistance to aminoglycoside antibiotics is the inactivation by their modifying enzymes. Among the aminoglycoside modifying enzymes, acetyltransferases, phosphotransferases and nucleotidyltransferases especially adenylyl transferases are clinically important, resistance-related enzymes. Aminoglycoside N-acetyltransferases (AACs) are divided into four major groups based on the position of acetylation: AAC(1), AAC(2′), AAC(3), and AAC(6′) [26,80]. Another novel type of aminoglycoside modifying enzyme, the enhanced intracellular survival (Eis) protein, was identified in *Mycobacterium tuberculosis* [81]. Unlike other AACs, Eis and its homologues acetylate multiple amino groups of aminoglycosides and are distributed in *Mycobacterium* and other *Actinobacteria* [82]. Moreover, Eis proteins are composed of more than 400 amino acid residues and, their amino acid sequences and the crystal structures are completely different from those of other AACs [83]. It is hypothesized that the antibiotic modifying enzymes in pathogenic bacteria were evolved from those in antibiotic-producing bacteria [7,8,9]. Here this hypothesis is re-evaluated from the point of amino acid sequence similarities of AACs, revealing that it is not necessarily true (Appendix A). The amino acid sequences of AAC(3)s in pathogenic bacteria such as AAC(3)-IIa (GB Nos. are in the parentheses; CAA31895), AAC(3)-IIIb (AAA25682), AAC(3)-IVa (CAA25642), AAC(3)-IIb (AAA26548), AAC(3)-IIc (CAA38525), AAC(3)-IIIa (CAA39184), AAC(3)-IIIc (AAA25683), and AAC(3)-VIa (AAA16194) are closely similar to those of aminoglycoside-producers (AAA88552, AAA26685, AAA25334, and BAA78619). For example, E values between AAA88552 (type VII AAC(3) from *S. rimosus* subsp. *paromomycinus*, a paromomycin-producer) and CAA31895, AAA25682, CAA25642, AAA26548, CAA38525, CAA39184, AAA25683, and AAA16194 are 2.9e-31, 1.4e-31, 1.8e-09, 1.2e-26, 1.1e-34, 1.3e-36, 6.8e-32, and 3.8e-24, respectively (Appendix A). However, the nucleotide sequence similarities (E values) between sequence of an aminoglycoside producer (AAA88552) and those of pathogenic bacteria (CAA31895 and AAA25682) are 11 and 0.42 (identities are 56.0% and 57.4%, and similarities are 66.1% and 66.9%), respectively, reflecting the different guanine+cytosine contents of these bacteria. On the other hand, those of the producers and of AAC(3)-Ia (AAO49599), AAC(3)-Ib (AAA88422), and AAC(3)-Ic (CAD53575) and AAC(3)-Id (AAR21614) are dissimilar (Appendix A). Instead, AAC(3)-Ia (AAO49599) shows a slight similarity to glucosamine-6-phosphate acetyltransferases from *Aspergillus fumigatus* (2VEZ-A, E = 5.4e-02) and *Trypanosoma brucei* (3I3G-B, E = 2.0e-02) (Appendix A); AAC(3)-Ib (AAA88422) shows a slight similarity to ribosomal protein S18-alanine N-acetyltransferase of *E. coli* (NP_418790, E value = 0.039), and N-α-acetyltransferase 20 isoform a of *H. sapiens* (NP_057184, E value = 0.098); and AAC(3)-Ic (CAD53575) shows a slightly similarity to acyl-CoA N-acyltransferase (NAT) superfamily protein of *Arabidopsis thaliana* (NP_001190321, E value = 2e-05), and N-acetyltransferase 8 of *Mus musculus* (NP_075944, E value = 0.010), suggesting that AAC(3)s in the pathogenic bacteria are evolutionally related not only to acetyltransferases of aminoglycoside-producing bacteria but also to other types of acetyltransferases of bacteria, fungi, plants and animals. Even between AAA88552 (AAC3 of a paromomycin-producer) and CAA31895 (AAC3 of plasmid pWP113a), similarity value E is 7.1 at their nucleotide sequence level instead of 2.9e-31 at their amino acid sequence level. As described above, the paromomycin-producer *S. rimosus* possesses two aminoglycoside acetyltransferase genes located far outside the biosynthetic gene cluster. One of them AAA88552/CAG44462 is AAC (3) and the amino acid sequence is very similar to other AAC enzymes from aminoglycoside producers, indicating that the amino acid sequence is not influenced by the location (Appendix A).

AAC(6)s in pathogenic bacteria form three clusters (B, D and E in Appendix A) in the phylogenetic tree constructed on the basis of their amino acid sequence homology. This result is in accord with that of Salipante and Hall [84]. While each member in the same cluster shows high sequence homology, that in the different clusters shows almost no homology at all. Furthermore, amino acid sequences in cluster B show relatively high homology to aminoglycoside N-acetyltransferase AAC(6′)-Ii from various species of *Enterococcus*; those in cluster D exhibit relatively high homology to acyl-CoA N-acyltransferase from *Arabidopsis thaliana* (NP_201544, E value = 0.003), diamine acetyltransferase 2 isoform 3 from *H. sapiens* (NP_597998, E value = 3e-04), and GNAT family N-acetyltransferase from *Microcystis aeruginosa* (WP_012265151, E value = 2e-04); and those in cluster E exhibit relatively high homology to those of acyl-CoA N-acetyltransferase from *Clostridioides difficile* (YP_001087683, E value = 0.018), spermidine/spermine acetyltransferase from *B. subtilis* (NP_390537, E value = 0.006), and lysine N-acetyltransferase from *M. tuberculosis* (YP_009358719, E value = 1e-06); suggesting again that the aminoglycoside acetyltransferases, AAC(6)s, in the pathogenic bacteria are evolutionally related not only to acetyltransferases of aminoglycoside-producing bacteria but also to other types of acetyltransferases of bacteria, plants and mammals. In other words, pathogenic bacteria had evolved the resistance genes from their divergent original proto-resistance genes and/or resistance-related and resistance-unrelated genes acquired through horizontal gene transfer [84,85]. The arrangement of the genes near AAC(3)-IId in *Pseudomonas aeruginosa* strain PA34 (WP_000557454) is completely different from that (CAH58703) in *S. fradiae*, a neomycin-producer (Appendix A). However, the identity (E value) of the amino acid sequences between WP_000557454 and CAH58703 is 3.1e-31, suggesting that these two are highly similar at their amino acid sequence level (Appendix A), although these two species are completely different in taxonomical classification. On the other hand, acetyltransferases from Actinobacterial species are divided into four groups: aminoglycoside 3-acetyltransferases, streptothricin-acetyltransferases, aminoglycoside 6-acetyltransferases, and Eis proteins. However, no amino acid sequence homology is observed between the members of each group (e.g., AAA88552, CAA51674, CAF60525, and NP_628362 in Appendix A, respectively).

Aminoglycoside phosphotransferases (APHs) catalyze the regiospecific transfer of the γ-phosphoryl group of ATP to one of the hydroxyl groups on aminoglycosides. They are divided into several groups: APH(4), APH(6), APH(9), APH(3′), APH(2”), APH(3”), and APH(7”)[80]. They form several clusters in the phylogenetic tree constructed on the basis of their amino acid sequences (Appendix A). Streptomycin phosphotransferase (BAG22761), spectinomycin phosphotransferases (ABW87797, AAB66655, and U70376_3) belong to cluster A. Although two spectinomycin phosphotransferases (ABW87797 and AAB66655) give high similarity with each other, these two give almost no similarity to another phosphotransferase (U70376_3) and streptomycin phosphotransferase (BAG22761) (Appendix A). In addition, these two show almost no similarity to the phosphotransferases from pathogenic bacteria (AAA26443, AE004612_1, and CAA25854). SmartBlast analyses suggest that SpcN from *S. spectabilis* (ABW87797) shows high similarities to phosphotransferases of *S. hygroscopicus* (WP_06628960, E = 1e-172), *S. aureocirculans* (WP_030559002, E = 3e-170), *S. silvensis* (WP_107450217, E = 1e-155), and *Thermobifida halotolerans* (WP_068692512, E = 5e-127), indicating that these phosphotransferases form one group in *Actinobacteria*. In contrast, the phosphotransferase from *S. netropsis* (U70376_3) shows similarity to hydroxyurea phosphotransferase from *S. pharetrae* (WP_086170844, E = 2-174), streptomycin 3”-phosphotransferase from *P. aeruginosa* PAO1 (NP_250549, E = 4.0e-15), and streptomycin 3-kinase from *Deinococcus radiodurans* R1 (NP_294178, E = 2.0e-08), indicating that these phosphotransferases form another group. On the other hand, the phosphotransferases of pathogenic bacteria in cluster B such as CAA23892, AE004828_7, and CAA24789 show high similarity scores not only to those of other pathogenic bacteria such as AAA2641, CAA23656, and AAA26442 but also to those from aminoglycoside-producing bacteria such as WP_063841674 (butirosin), CAN38351 (sisomicin), CAF34039 (gentamicin), CAG44623 (paromomycin), CAG34043 (ribostamycin), CAH58684 (neomycin), and others (Appendix A). They are aminoglycoside 3′ or 3”-phosphotransferases and are derived both from Gram-positive and Gram-negative bacteria. The phosphotransferases in cluster C exhibit the high similarity to members within the cluster, but not to those in other clusters (Appendix A), supporting the concept that at least some resistance-involved phosphotransferases in pathogenic bacteria are only far distantly related to those in aminoglycoside-producing bacteria. The members in cluster C are aminoglycoside 2”-phosphotransferases and stem from Gram-positive bacteria. No report has been published on the involvement of nucleotidyltransferase in the self-resistance in *Streptomyces* [80,86], although lincosaminide and muraymycin nucleotidyltransferases have been reported [87,88].

The modification of the ribosomal target is another mechanism of resistance, that is, the methylation of 16S ribosomal RNA by methyltransferases. Depending on the modified nucleotide position at the A-site of 16S rRNA, the methyltransferases involved in aminoglycoside resistance are classified into two groups: N7-G1405 (methylation of N7 position of guanine-1405) 16S rRNA methyltransferases and N1-A1408 (methylation of N1 position of adenine-1408) 16S rRNA methyltransferases [89]. A phylogenetic tree constructed on the basis of their amino acid sequences is shown in Appendix A. It is composed of three clusters: clusters A, B, and C. Members in cluster A belong to N1-A1408 methyltransferases, and those in clusters B and C are comprised of N7-G1405 methyltransferases. It should be pointed out that the members in cluster B from aminoglycoside-producers and those in cluster C from plasmids in pathogenic bacteria are closely related phylogenetically, while those in cluster A are dissimilar (Appendix A). The genes for rRNA methyltransferases in cluster C are present on plasmids isolated from Gram-negative bacteria. Cluster A includes rRNA methyltransferases both from *Actinobacteria* and a plasmid in Gram-negative bacteria (Appendix A). SpcM from *S. spectabilis* (spectinomycin-producer, ABW87807) was proposed to be an rRNA methyltransferase [57]. However the amino acid sequence is completely different from those of other rRNA methyltransferases (Appendix A). Instead, it shows high similarity scores with N6-cytosine_N4-adenine site-specific DNA methyltransferases from *Streptoalloteichus hindustanus* (WP_073483148, E = 1e-103), *Photorhabdus laumondii* (WP_113024414, E = 3e-97), and from *Erwinia toletana* (WP_017800138, 2e-95). VldO from *S. hygroscopicus* subsp. *limoneus* (ABC67279, validamycin-producer) was predicted to be an O-methyltransferase [69]. The amino acid sequence is completely different from other rRNA methyltransferases (Appendix A). Alternately, it is more similar to predicted O-methyltransferase YrrMs from *S. corchorusii* (WP_079082699), *S. leeuwenhoekii* (CQR66120), and *Amycolatopsis sulphurea* (WP_098513546). So, it is doubtful whether it functions as a self-defender. Summarizing these data, it is concluded that the heterogeneity of 16S rRNA methyltransferases involved in the drug resistance is less than those of the acetyltransferases and phosphotransferases. These are the results of analyses at their amino acid sequence level. However, it should reevaluate the evolutional relation of the resistance genes in the antibiotic producers and the pathogens at the nucleotide sequence level. For example, even between CAE46946 (Kmr of a kanamycin producer) and AAN87711 (methyltransferase NbrB of *Citrobacter freundii*), similarity value E is 0.79 at their nucleotide sequence level instead of 9.0e-21 at their amino acid sequence level.

To accomplish high-level resistance, pathogenic bacteria should accumulate a small but distinct increase in environmental adaptation by sequential mutations. Before achievement of such level of resistance, bacterial cells use efflux pumps or efflux transporters as the first line of defense against antibiotics. The antibiotic transporters are divided into five families: the small multidrug resistance (SMR) family, the multidrug and toxic compound extrusion (MATE) family, the major facilitator superfamily (MFS), the ATP-binding cassette (ABC) family and the resistance-nodulation-cell division (RND) family [90,91]. These transporters function as a network, that is, more than one transporters are involved in the exclusion of one xenobiotic such as antibiotics and pollutants [92,93]. Therefore, the antibiotic efflux transporters are only a part of an overall detoxifying system consisting of a large range of coordinated membrane proteins. This is reflected in the diversity of the transporters implicated in the aminoglycoside self-resistance. Appendix A shows a phylogenetic tree constructed on the basis of the amino acid sequences of the transporters implicated in the aminoglycoside self-resistance (Appendix A). The tree is divided into four clusters: cluster A, cluster B, cluster C, and cluster D. The cluster C is further divided into four sub-clusters: C1, C2, C3, and C4. The sub-cluster C4 contains most of the aminoglycoside-related transporters constituting of 16 members. The similarity values Es between SAV1866 in *Staphylococcus aureus* (WP_124781844) and one of those in sub-cluster C4 range from 4.6e-06 (*S. hygroscopicus* Hyg28, ABC42565) to 4.5e-70 (*S. tenebrasius* AprW, CAF33030). The other sub-clusters contain ABC permeases such as *S. glaucescens* GacX, *S. kasugaensis* KasL, and *S. kasugaensis* KasM, EamA-like MFS proteins such as *S. tenebrarius* TobU. The cluster B are ABC transporter components with low similarity to SAV1866 in *S. aureus* (WP_124781844) (Appendix A). The cluster D includes 9 MFS members. The nine members in the cluster D and 2 members in the cluster A (*M. olivasterospora* ForV, CAF31538 and *M. inyonensis* Sis26, ACN38360) are clearly divided into two groups (Appendix A). The members of the group A belong to drug: H^+^ antiporter 14-spanner (DHA14) or drug: H^+^ antiporter 12-spanner (DHA12) drug efflux family and those of the group B belong to cyanate permease (CP) family [94]. Together with the fact that the cluster A includes FHA-domain containing proteins (*S. tenjimariensis* SteF24.27c, CAH60152 and *M. inyonensis* Sis27, ACN38361), it is suggested that the aminoglycoside-related transporters in the producing and pathogenic bacteria are composed of tremendously various proteins. Whereas the sequence similarities between transporters in the aminoglycoside producing and pathogenic bacteria are very high in ABC transporters described above, those among MFS members are very low (Appendix A). Hence, the similarity values Es between *E. coli* CynX (AAB18065) and one member in group B in Appendix A span from 1.5 (*S. tenebrarius* TobT, CAH18551) to over 1e+03 at the amino acid sequence level. Recently, the alteration of ribosomal targets by 16S rRNA methyltransferases was reported to confer resistance to most aminoglycosides in pathogenic bacteria. This type of resistance was not previously thought to be a clinically relevant mechanism of resistance [95,96].

### 2.2. Macrolide and Related Antibiotics

Macrolide antibiotics also belong to the protein synthesis inhibitors, and are divided into 12-, 14-, 15-, 16-, and 18-membered ring groups based on the chemical structures of the number of atoms in the macrocyclic lactone ring [97,98]. However, 14-membered (erythromycin, oleandomycin, narbomycin, and others) and 16-membered (tylosin, carbomycin, spiramycin, and others) ring macrolide antibiotics are major antimicrobials. The macrolide antibiotics bind to the ribosomal nascent peptide exit tunnel (PNET) adjacent to the peptidyl transferase center (PTC), and prevent protein biosynthesis. However, the binding mode is controlled discretely in the molecular-species-specific manner [19,22]. The macrolide antibiotics show antimicrobial activity against both Gram-positive and some Gram-negative bacteria. In addition, they are active against *Mycoplasma*, *Chlamydia*, *Legionella*, and *Coxiella*. Moreover, some macrolide antibiotics function as motilin receptor agonists [99].

Methymycin is a 12-membered ring macrolide antibiotic produced by *S. venezuelae*. The methymycin biosynthetic gene cluster was cloned and sequenced [100,101]. It contains genes for two rRNA methyltransferases (*pikR1* and *pikR2*) and β-glycosyltransferase (*desR*) which are involved in the self-resistance. These genes are also implicated in the self-resistance against pikromycin and narbomycin, 14-membered ring macrolide antibiotics coproduced by *S. venezuelae*. The parts of the biosyntheses share a common route in these three antibiotics. Interestingly, the presence of macrolide glycosyltransferases has been reported in a number of macrolide non-producers such as *S*. *vendargensis* and *S*. *lividans* [102,103]. However, the detailed function of these glycosyltransferases has not been defined. Erythromycin is the best known member of the 14-membered group and was isolated from *Saccharopolyspora erythraea*. The biosynthetic gene clusters were cloned from *S*. *erythraea* [104,105,106]. The 23S rRNA methyltransferase gene (*ermE*, SACE_0733) is present within the cluster. In addition, 11 further rRNA methyltransferases are present in the genome. Furthermore, two putative macrolide glycosyltransferases (SACE_1884, and SACE_3599), and a number of efflux proteins for antibiotics exist outside of the cluster (Appendix A) [105]. These genes may be involved in the self-resistance. Oleandomycin is a 14-membered ring macrolide antibiotic isolated from *S. antibioticus*. Glycosyltransferases (OleD, Oleg1, OleG2, and OleI) and ABC transporters (OleB and OleC) were proposed to be implicated in the self-resistance [107,108,109,110]. As described above, pikromycin and narbomycin are 14-membered macrolide antibiotics produced by *S. venezuelae*, and the parts of the biosynthetic routes and the resistance genes share with those of methymycin [111]. Lankamycin is a 14-membered macrolide antibiotic. Interestingly, the biosynthetic genes are located on the giant linear plasmid pSLA2-L in *S. rochei* together with those of lankacidin, a 17-membered carbocyclic polyketide compound [112]. Two ABC transporter genes (*lkcI* and *lkcJ*) are present within the biosynthetic gene cluster, and another ABC transporter gene (*lkmN*) is present outside of the cluster. In addition, two efflux transporter genes (ST1928_p012 and ST1928_p024) exist outside of the cluster. These transporters are involved in the self-resistance (GB No.NC_004808). Tylosin is a 16-membered ring macrolide antibiotic developed for veterinary use [98]. Four self-resistance determinants are defined. Among them, three (*tlrB*, *tlrD*, and *tlrC*) are within the biosynthetic gene cluster, and one (*tylA*) exists outside of the cluster. *tlrA*, *tlrB*, and *tlrD* code for rRNA methyltransferases, and *tlrC* encodes an efflux protein [113,114,115,116]. The spiramycin biosynthetic gene cluster was cloned and sequenced [117] (GB No.CP012382). Two ABC transporters (SrmB and DrrA), two rRNA methyltransferase (SrmD and SrmA), and one macrolide glycosyltransferase (MgtA/GimA) are involved in the self-resistance. *mgtA/gimA* gene is present immediately downstream of *srmA* (GB No.SAM23877_5808) outside of the biosynthetic gene cluster [118]. Carbomycin is a 16-membered macrolide antibiotic isolated originally from *S. halstedii* and the biosynthetic gene cluster was cloned from *S. thermotolerans* (GB No.KR818745). Two transporter genes (*carA*/*cbm25* and *cbm26*) and rRNA methyltransferase gene (*carB*/*cbm7*) were identified as resistance determinants [119,120]. Interestingly, amino acid sequences of CarA from *S. thermotolerans*, TlrC from *S. fradiae* and SrmB from *S. ambofaciens* are closely similar (similarity values Es = 3.4e-85~1.1e-115). Mycinamicins are 16-membered macrolides constituting the mycinamicin subgroup produced by *Micromonospora griseorubida*. The biosynthetic gene cluster was cloned [121] (GB No.AB089954). MyrB, an rRNA methyltransferase, was reported to be implicated in the self-resistance [122]. Tiacumicin B is an 18-membered ring macrolide produced by *Dactylosporangium aurantiacum*. The gene cluster was cloned, and four transporter genes were reported within the cluster [123] (GB No.HQ011923).

Lincomycin is a member of lincosamide group antibiotics consisting of amino acid and sugar moieties. It is a protein synthesis inhibitor [124,125]. Three resistance-related genes, *lmrA* and *lmrC* encoding efflux pumps and *lmrB* encoding rRNA methyltransferase were identified within the biosynthetic gene cluster [126,127,128]. A lincosamide resistance determinant *clr*, a 23S rRNA methyltransferase gene, was also isolated from *S. caelestis*, a celesticetin-producer [129]. Hormaomycin is a bacterial signaling metabolite with narrow-spectrum antibiotic activity produced by *S*. *griseoflavus*. Pyrrolobenzodiazepine derivatives, tomaymycin, anthramycin and siberomycin, are sequence-selective DNA alkylating agents. Lincomycin, hormaomycin, tomaymycin, anthramycin and siberomycin are known to be derived from a common intermediate (3-vinyl-2,3-pyrroline-5-carboxylic acid), and to constitute similar biosynthetic gene clusters [130]. The biosynthetic gene cluster for hormaomycin in *S. griseoflavus* contains two transporter genes, *hrmU* and *hrmV* [131]. The tomaymycin biosynthetic gene cluster was cloned as 26kb DNA fragment, and one transporter gene, *tomM*, was identified within the cluster [132] (GB No. FJ768957). The biosynthetic gene cluster of anthramycin was cloned and sequenced. Adjacent genes *orf9* and *orf10* were proposed to encode transporters. From the amino acid sequence similarity of Orf8 to UvrA and DrrC, it is suggested a role as a transporter or the excision nucleotide repair system in the resistance [133] (GB No. EU195114). The siberomycin biosynthetic gene cluster was cloned as 32.7kb DNA fragment. One transporter gene *sibF* was identified within the cluster [134] (GB No. FJ768674).

Streptogramins/pristinamycins are a family of antibiotics that are composed of a mixture of two chemically different compounds: group A streptogramins/pristinamycin II constituting of poly-saturated macrolactones, and group B streptogramins/pristinamycin I constituting of cyclic hexadepsipeptides. Both group A and group B streptogramins are protein biosynthesis inhibitors, and function synergistically to provide greatly enhanced levels of antimicrobial activity. Pristinamycin I and pristinamycin II biosynthetic gene clusters were cloned as a 210kb DNA fragment interspersed to two segments by the insertion of 90kb cryptic secondary metabolite cluster [135]. Two transporter resistance genes, *snbR* and *ptr*, were identified, one within the cluster and another outside of the cluster [136]. Virginiamycin M and virginiamycin S belong to group A and group B streptogramin families, respectively. A part of the biosynthetic gene cluster was cloned [137]. Three transporter genes are present within the cluster. In addition, virginiamycin M was reported to be stereospecifically reduced to an inactive derivative in *S. virginiae* [138]. Griseoviridin and viridogrisein/etamycin belong to group A and group B streptogramin antibiotics produced by *S. griseoviridis*, respectively. Three transporter genes (*sgvT1*, *sgvT2*, and *sgvT3*) are found within the biosynthetic gene cluster [139] (JX508597). Evernimicin and avilamycin are orthosomycin group antibiotics inhibiting protein biosynthesis. However, cryo-electron microscopical analyses of the complexes of orthosomycins and *E. coli* ribosome revealed that the binding site on the large subunit is different from that of other antibiotics such as macrolides and thiostrepton [140]. The biosynthetic gene cluster for evernimicin was cloned. Two 23S rRNA methyltransferase genes (*evrH* and *orf6*) and two efflux pump genes (*evrE* and *evbB*) were identified within the cluster [141]. These may be involved in the self-resistance. Avilamycin biosynthetic gene cluster was cloned as 60kb fragment from *S. viridochromogenes*. Two rRNA methyltransferases (AviRa and AviRb) and two antibiotic transporters (AviABC1 and AviABC2) were clarified [142] (GB No. AF333038). Viomycin and capreomycin are tuberactinomycin group antibiotics used for the treatment of multi-resistant tuberculosis. They inhibit bacterial protein biosynthesis by blocking translocation of the mRNA-tRNA complex. The biosynthetic gene cluster for viomycin was cloned. Viomycin phospho-transferase (Vph) and a permease (VioE) are assumed to be involved in the self-resistance [143] (GB No. AY263398). The capreomycin biosynthetic gene cluster was cloned. Within the cluster, rRNA methyltransferase gene (*cmnU*) and capreomycin phosphotransferase gene (*cph*) were detected [144] (EF472579). In addition, capreomycin acetyltransferase (Cac) was proposed to be implicated in the self-resistance [145].

In summary, in contrast to the cases of aminoglycosides the major resistance mechanisms to macrolide and related antibiotics in producing bacteria are efflux of the drugs and methylation of rRNA. As described above, the use of transporters/efflux pumps is the first line of defense against antibiotics by decreasing the intracellular level of antibiotics before the cell activates the various other tools of defense [146]. This is true not only in pathogenic bacteria but also in antibiotic-producing bacteria, so there are many genes encoding transporters/efflux pumps in the genomes, although the genes for drug transporters/efflux pumps have not been detected within the biosynthetic gene clusters of macrolide antibiotics methymycin and mycinamicin. These transporters/efflux pumps may be involved in the first line of defense in the producing bacteria. On the other hand, there are some questions. What is the real role of the assumed resistance-related proteins such as the glycosyltransferase (DesR) in methymycin-producer [101], the glycosyltransferases (OleG1, OleG2) in oleandomycin-producer [109,110], the phosphotransferase (Vph) in viomycin-producer [143], the phosphotransferase (Cph) and the acetyltransferase (Cac) in capreomycin-producer [144,145]? Cac is supposed to be an aminotransferase, as the amino acid sequences are highly similar to many aminotransferases from *Actinobacteria*. How did rRNA methyltransferases become resistance tools in antibiotic producers? How were the resistance-related these proteins evolved? Blast analyses revealed that amino acid sequence of DesR is similar not only to glycosyltransferases from various *Streptomyces* species but also to the glycosyl hydrolase from *Schizosaccharomyces pombe* (NP_595060, E value = 6e-78), and β-xylosidase from *Arabidopsis thaliana* (NP_196535, E value = 1e-38); that of Vph is similar not only to viomycin phosphotransferases from various species but also to aminoglycoside phosphotransferases from *Seinonella peptonophila* (WP_073150348, E value = 8e-24) and other Firmicutes bacteria; and that of Cac is similar not only to aminotransferases from various *Actinobacteria* but also to SufS family cysteine desulfurases from *Microcystis aeruginosa* (WP_002796790, E value = 2e-19) and other bacteria and plants. Together with the fact that similar functional proteins distribute in a wide range of phyla/biosphere irrespective of the large difference of GC contents in genomes, e.g., Erm-like protein is present in *Yuhushiella deserti* (GB No. SFO87742, E value = 3e-130) as well as *Homo sapiens* (GB No.NP_001335005, E value = 4e-19), these results indicate that the antibiotic producing bacteria have evolved the resistant systems from the accidentally acquired related genes within the biosynthetic gene clusters. This has happened in a long evolutional history of life by overcoming the high barrier of GC contents in the genomes [147].

The resistance mechanism to macrolide and related antibiotics in pathogenic bacteria, on the other hand, is divided into mutations and modifications of 23S rRNA, macrolide efflux systems, macrolide inactivation by phosphotransferases and esterases, and others. Macrolides interact primarily with A2058 and A2059 of the 23S rRNA, and the mutations in these nucleotides have been found in many macrolide-resistant pathogenic bacteria, such as *Mycobacterium*, and *Helicobacter*. Mutations at G2057 and C2611, sometimes in combination with A2058 or A2059, have been detected in *Streptococcus*, and *Staphylococcus*. In addition, mutations in genes encoding ribosomal proteins L4 and L22, which contact macrolides in the ribosome-macrolide complexes [148], confer resistance to macrolides in various pathogenic bacteria [149]. This type of mutations includes not only amino acid exchange, but also deletion and insertion. The resistance mechanisms of these types have not been observed in the macrolide-related antibiotic producers. The second type of resistant mechanism is the rRNA modification by rRNA methyltransferases encoded by *erm* genes. *erm* genes encode the methyltransferases which methylate A2058 located in the peptide exit tunnel of rRNA. This type of mechanism confers resistance to 14-, 15-, and 16-membered macrolides and ketolides, as well as to lincosamides and streptogramin B. Now, over 40 *erm* genes have been reported [149]. Interestingly, the amino acid sequence of Erm in *Stretococcus pneumoniae* (GB No.BBG37057) is very similar to that in CarB in *S. thermotolerans* (GB No.AAC32026, E value = 1.8e-14) and PikR2 in *S. venezuelae* (GB No. AAC69327, E value = 1.1e-19) and methyltransferases in other macrolide producers. The third type of mechanism is related to macrolide efflux systems encoded by *mef*, *msr*/*mel* and *lsa* genes. E values at nucleotide sequence level are 0.05 (*erm*/*carB*), and 3.8e-3 (*erm*/*pikR2*), respectively. Coexpression of *mef* and *msr* is required for high level macrolide efflux, and these proteins interact synergistically to increase macrolide resistance. However, the detailed mechanism remains to be clarified, although it was reported that these proteins form a composite efflux pump [150]. While the amino acid sequence of Msr (GB No.WP_053875754) in *S. aureus* is similar to that of Hyg28 in *S. hygroscopicus* (E value = 8.9e-30), that of MefS in *S. pneumoniae* is not similar to any sequences in macrolide producers analyzed. It is possible that some other transporters outside of the biosynthetic gene clusters in producers are similar to those in the pathogenic bacteria. The fourth mechanism is macrolide inactivation by phosphoesterases and esterases that are encoded by *mph* and *ere* genes, respectively. Macrolide phosphotransferases are macrolide-inactivating enzymes widespread in Gram-positive and Gram-negative bacteria, and belong to the same family as aminoglycoside phosphotransferases and protein kinases [151]. Macrolide phosphotransferases confer resistance to a wide variety of macrolide antibiotics. However, details remain to be elucidated [152]. As for macrolide esterases that inhibit the antimicrobial activity of macrolides, five families are reported [152]. These enzymes are thought to be originally diverse, although they provide macrolide resistance to pathogenic bacteria [153]. This type of resistance has not been reported in macrolide producing bacteria. In summary, rRNA methyltransferases and phosphotransferases in pathogenic bacteria are closely related to those in macrolide-producing bacteria, whereas rRNA mutations and efflux pumps are scarcely related with each other. These resistance characters may have been transferred from other sources.

### 2.3. Tetracycline and Chloramphenicol

Tetracyclines have been used for the treatment of a wide variety of Gram-positive and Gram-negative bacterial infections and for animal feeds and aquaculture since the 1940s. Now, third and fourth generation compounds have rejuvenated clinical prospects for this drug class. Tetracyclines inhibit bacterial protein biosynthesis by binding to the 16S rRNA, preventing the delivery of tRNA to the A-site [19,23]. The therapeutic potential in cardiovascular diseases was also reported, as tetracyclines inhibits matrix metalloproteinases [154]. The oxytetracycline biosynthetic gene cluster was cloned [155,156]. A total of 21 ORFs were clustered between two resistance genes *otrA* and *otrB*, encoding a ribosomal protection protein (RPP) [157] and a transporter, respectively [158]. The chlortetracycline biosynthetic gene cluster was cloned from *Kitasatospora aureofaciens* [159,160] (HM627755). Within the cluster, one ribosomal protection protein gene (*ctcC*), and three transporter genes (*ctcR*, *ctcY* and *ctc2*) are detected. Recently, Forsberg et al. reported a novel family of tetracycline-inactivating enzymes by soil functional metagenomic selections, although the exact function in the drug resistance of pathogenic bacteria is not clear [161].

Chloramphenicol is an antibiotic produced by *S*. *venezuelae* and other *Streptomyces* species. It inhibits protein biosynthesis by interacting with the bacterial 50S subunit of ribosome and blocking amino acyl-tRNA binding at the A-site, and is used for the treatment of Gram-positive and Gram-negative bacterial infection. However, the side effects such as bone marrow suppression, nausea and diarrhea hamper its common use. The chloramphenicol biosynthetic gene cluster was cloned and sequenced [162,163] (NC_018750). One gene for MFS efflux pump (*cmlF*, SVEN_RS04435) is present within the cluster, and another gene for MFS transporter (*cmlV*, SVEN_RS20160) is present outside of the cluster. Interestingly, chloramphenicol phosphotransferase gene (*cpt*, SVEN_RS20155) is present just adjacent to *cmlV*. Acetylchloramphenicol was proposed to be an intermediate in chloramphenicol biosynthesis [164], although the exact role of acetylation of chloramphenicol in the self-resistance has not been elucidated. Chloramphenicol hydrolase was also proposed to be involved in the self-resistance [165].

As for the resistance to tetracyclines in pathogenic bacteria, at least four mechanisms have been reported, that is, binding site mutations, ribosomal protection proteins, efflux pumps, and enzymatic inactivation [166,167]. Because most bacteria have multiple rRNA copies, mutations in rRNA conferring tetracycline resistance are usually found in bacteria with low rRNA copy numbers, such as *Propionibacterium acnes*, *Helicobacter pylori*, *Mycoplasma bovis*, and *S. pneumoniae*. For example, *S. pneumoniae* with mutations C1054T and T1062G/A in 16S RNA is resistant to tigecycline when four genomic copies of 16S rRNA are mutated. Mutations in the *rpsJ* encoding the 30S ribosomal subunit protein S10 are also reported to confer resistance to tetracyclines in *S. pneumoniae* [168]. Mutations in *rpsJ* are described in various clinical isolates of Gram-negative bacteria. Furthermore, the same authors described nonsense mutations in *spr1784* encoding a 16S rRNA methyltransferase resulting in tetracycline resistance in *S. pneumoniae* [168]. Tetracycline ribosomal protection proteins (RPPs) are GTPases with significant sequence and structural similarity to elongation factors EF-G and EF-Tu. They are found both in Gram-positive and Gram-negative bacteria. The most common RPPs are TetO and TetM. The sequence similarity values between OtrA from tetracycline producer *S. rimosus* (GB No.ALS03934) and TetM from *E. faecalis* (GB No.CAA63530), and TetO from *Campylobacter jejuni* (AAA23033) are 5e-79 and 2.2e-80 at the amino acid sequence level, and 2.1e-38 and 3.7e-47 at the nucleotide sequence level, respectively, indicating that these proteins are closely related each other. It is assumed that RPPs like TetM and TetO have originated in the tetracycline producers. The most common tetracycline-specific efflux pumps are members of the major facilitator superfamily transporters. They are classified in seven groups. The group 1 pumps such as TetA and TetB possess 12 transmembrane segments and distribute mostly in Gram-negative bacteria. The group 2 pumps like TetK and TetL contain 14 transmembrane segments and are present mostly in Gram-positive bacteria [91]. The amino acids of TetK and TetL from Gram-positive bacteria have some sequence similarity to those of OtrB, CtcR, and Ctc2 from tetracycline producers, showing that the similarity values Es are in the range of 1e-05 to 1e-09. Enzymes capable of inactivating tetracyclines are rare compared with enzymes that inactivate other antibiotics. Three enzymes are known to be implicated in the tetracycline inactivation: flavin-dependent monooxygenases encoded by *tetX* family genes, NADP-requiring tetracycline modifying enzymes, and xanthine-guanine phosphoribosyltransferases [161]. These enzymes have not been described in tetracycline producing bacteria, although they may possess their dissemination potential into the clinic near future.

The resistance mechanisms to chloramphenicol in pathogenic bacteria are due to the enzymatic modification of chloramphenicol, efflux pumps, and target modifications. The amino acid sequence of the acetyltransferase from *Streptomyces acrimycini* (GB No.CAT_STRAC) is extremely similar to that from *Haemophilus influenzae* (GB No. CAA37806, E = 9.5e-40), to that from *Shigella flexneri* plasmid (GB No. CAA30695, E = 3e-40), and that from *S. aureus* plasmid (GB No. CAA26367, E = 2.3e-35). However, whether the gene for this type of acetyltransferase is present in chloramphenicol producing *Streptomyces* species is not clear, although acetylchloramphenicol was detected in chloramphenicol producing *S. venezuelae* [164]. Whereas a chloramphenicol phosphotransferase is detected in chloramphenicol producing *S. venezuelae*, this type of phosphotransferase has not been reported in pathogenic bacteria. Similarly, a chloramphenicol hydrolase gene is found in the chloramphenicol producer *S. venezuelae*, but not in pathogenic bacteria [169]. Related to the chloramphenicol resistance, a number of efflux pumps have been reported [24]. The amino acid sequence of CmlV (GB No.AAB36568) from chloramphenicol producing *S. venezuelae* is similar to CmxB from *Corynebacterium striatum* plasmid (GB No. AAG03380, E = 1.4e-23), and to FexA from *Staphylococcus lentus* plasmid (GB No. CAD70268, E = 9.6e-02), but not to CmlA from *Salmonella typhimurium* plasmid (GB No. CAD31707, E = 2.2) and to Cml from *E. coli* plasmid (GB No. AAA26079, E = 4.3e+02). On the other hand, the *cfr* gene encoding an rRNA methyltransferase that targets A2503 in the domain V of the 23S RNA has been identified on a number of plasmids in *S. aureus* and other Gram-positive bacteria. Furthermore, the *cfr* gene was also found in chromosomal DNAs or on plasmids in some Gram-negative bacteria [170]. This type of rRNA methyltransferase-mediated chloramphenicol-resistance has not been described in chloramphenicol producing *S*. *venezuelae*.

### 2.4. Other Protein Synthesis Inhibitors

Kirromycin is a complex linear polyketide peptide-bonded to sugar-like moiety produced by *S. collinus, S. ramocissimus*, *S. cinnamoneus*, and *Nocardia lactamdurans*. Among the kirromycin-producers, *S. collinus* Tue365 and *S. ramocissimus* express kirromycin-sensitive elongation factors even during the kirromycin producing period, whereas *S. cinnamoneus* and *N. lactamdurans* encode kirromycin-resistant elongation factor [171,172]. EF-Tu3 from *S. coelicolor* A3(2), a kirromycin-non-producer, is also resistant to kirromycin [172]. The kirromycin biosynthetic gene cluster was isolated as 130kb DNA fragment containing 57 ORFs. Two MFS type transporters are detected within the cluster [173,174] (GB No.AM746336). Kirromycin shows strong antibacterial activity against *Streptococci*, some *Enterococci*, *Neisseria*, and *Haemophilus*, but not to *S. aureus*. The narrow antibiotic spectrum of kirromycin is explained by the sophisticated structural difference of EF-Tus in bacteria [175,176]. The role of kirromycin has not been clarified in pathogenic bacteria.

Bicyclomycin is a 2,5-diketopiperazine derivative and a selective inhibitor of the transcription termination factor Rho. It is isolated from *S. cinnamoneus* and shows a broad-spectrum antibiotic activity against Gram-negative bacteria. The biosynthetic gene cluster was cloned from *S. cinnamoneus*, but putative bicyclomycin gene clusters are bound to at least seven spanning *Actinobacteria* and *Proteobacteria*. The MFS type transporter gene *bcmT* is present within the cluster [177,178]. In a pandemic *P. aeruginosa* clone, the integrative and conjugative element carrying the metallo-β-lactamase gene and bicyclomycin resistance gene *bcr1* was reported [179]. *bcr1* encodes a efflux pump for bicyclomycin [180].

Thiostrepton is a thiopeptide group antibiotic isolated from *S. azureus* more than 50 years ago. This type of antibiotics inhibits protein synthesis by targeting the ribosome or ribosome-associated factors. They are active against clinically relevant methicillin-resistant *S. aureus* (MRSA), *E. faecium* (MREF), penicillin-resistant *S. pneumoniae* (PRSP) and vancomycin-resistant Enterococci (VRE). However, it is only limitedly applied clinically due to its poor solubility and toxicity. The biosynthetic gene clusters for thiostrepton and related antibiotics GE2270 and thiomuracin were cloned from *S. laurentii* and *Nonomuraea*, respectively [181,182] (GB Nos. FJ652572; FJ461359; FJ461360). An rRNA methyltransferase gene, that imparts self-resistance to thiostrepton in S. *laurentii*, is not linked to the biosynthetic gene cluster but instead it is located within a cluster of ribosomal protein operons [183]. Spontaneous thiostrepton-resistant mutants were isolated from *Thermus thermophilus*. The mutations were found in the L11-binding site of 23S rRNA [184]. Microccin P1 is a thiopeptide group antibiotic having a 26-membered macrocycle like thiostrepton. The microccin P1 biosynthetic gene cluster was isolated from *S. epidermidis* and compared with that of thiocillin from *B. cereus* [185,186,187]. The cluster for microccin P1 contains *tclQ* that encodes TclQ protein incorporating into the ribosome in place of L11 and conferring the self-resistance to microccin P1 [185]. The cluster for thiocillin contains two L11-like proteins TclQ and TclT that may be involved in the self-resistance [187]. The resistance mechanisms in this class of antibiotics are generally similar between producers and pathogens, although the details have not been elucidated yet.

## 3. Cell Wall/Membrane Synthesis Inhibitors

### 3.1. β-Lactams

β-Lactam group antibiotics including semi-synthetic penicillins and cephalosporins are the most commonly used antibiotics in the clinic for the treatment of Gram-positive as well as Gram-negative bacterial infections, although they have been used for almost one century. They are classified into five groups according to their chemical structures: penicillins, cephalosporins/cephamycins, clavulanic acid, thienamycin, nocardicin A and sulfazecin. Penicillin was isolated in 1929 as the first antibiotic by Fleming [188], and rediscovered in 1940 by Chain et al. [189] and in 1941 by Abraham et al. [190]. Penicillins and cephalosporins/cephamycins are produced by bacteria as well as fungi, whereas other β-lactam antibiotics are produced by bacteria [191,192,193]. The gene clusters for the biosyntheses of penicillins/cephalosporins/cephamycins were cloned from *Streptomyces clavuligerus* [194,195] (GB No. CM000913, SCLAV_4179~SCLAV_4214), *S. cattleya* [196] (GB No. NC_016111, SCAT_5676 ~ SCAT_5692), *Nocardia lactamdurans* [197,198], *Lysobacter lactamgenus* [199] (GB No. X56660), *Penicillium chrysogenum* [200] (GB No.CM002799, EN45_082610~EN45_082630), and *Aspergillus* (*Emericella*) *nidulans* [201]. Interestingly, whereas the genes for penicillin-binding proteins and β-lactamases, which are involved in the self-resistance in bacteria [5,202,203], are present within these clusters of bacteria, they are absent in those of the fungi, indicating strongly that penicillin-binding proteins and β-lactamases especially the formers are involved in the self-resistance [5]. Clavulanic acid is a potent inhibitor of a various kind of β-lactamases from pathogenic bacteria with antibacterial activity and was isolated from *S. clavuligerus* [204,205]. It has been used in combination with β-lactam antibiotics [206]. The gene cluster for the biosynthesis of clavulanic acid is located between cephamycin gene cluster and penicillin-binding protein and β-lactamase genes [195,207,208]. Comparison of the gene clusters for the biosynthesis of cephamycin in *S. clavuligerus* and *S. cattleya*, a clavulanic acid-non-producer, indicates that the clavulanic acid gene cluster is inserted between the cephamycin gene cluster and penicillin-binding protein/β-lactamase genes without affecting the presence of penicillin-binding protein and β-lactamase genes, suggesting that the penicillin-binding proteins and the β-lactamases play important roles in the protection from cephamycin but not from clavulanic acid in the producers [209]. However, precise role of penicillin-binding protein and β-lactamase genes in clavulanic acid biosynthesis remains to be elucidated [210].

Thienamycin displays antimicrobial activity against Gram-negative, Gram-positive as well as anaerobic bacteria. Unfortunately, however, it is extremely unstable in aqueous solution [211]. Consequently, using thienamycin as a progenitor natural product a various carbapenem compounds such as imipenem, meropenem and others with broad antibacterial activity have been synthesized and introduced to clinic [212]. The gene cluster for the biosynthesis of thienamycin is located on the plasmid of *S. cattleya* [213,214](GB No. AJ421798). Three genes, *thnF*, *thnJ*, and *thnS*, encoding N-acetyltransferase, transporter protein and β-lactamase, respectively, may be involved in the self-resistance [213]. In addition, *thnC* encoding efflux pump may be implicated in the resistance. Like other β-lactam antibiotics, thienamycin binds to penicillin-binding proteins (PBPs) of *E. coli*, especially to PBP2 [215], although the genes for the PBPs are not detectable within the thienamycin biosynthetic gene cluster. Alternatively, two PBP genes are present within the cephamycin biosynthetic gene cluster as described above. Nocardicin A is a monocyclic β-lactam antibiotic monobactam, and was isolated from *Nocardia uniformis* [216] and other actinomycetes. It shows moderate antimicrobial activity against Gram-negative bacteria and exhibits some β-lactamase resistance [217]. The monobactam antibiotic was later developed to clinically important aztreonam. The biosynthetic gene cluster of nocardicin A was cloned [218] (GB No. AY541063). Acetyltransferase (NocD) and transporter protein (NocH) were proposed to be involved in the self-resistance. Another monobactam antibiotic sulfazecin was isolated from *Pseudomonas acidophila*. It is active against Gram-negative bacteria [219] and is not inactivated by metallo-β-lactamases that make bacteria resistant to extended-spectrum β-lactam antibiotics. The gene cluster contains several transporter genes, a β-lactamase gene, and the multidrug transporter gene *mdtB* which may be involved in the self-resistance [220] (GB No. KX757706). However, the exact role of β-lactamase remains to be clarified.

In pathogenic bacteria, the predominant resistance mechanism of Gram-positive bacteria to β-lactam antibiotics is different from that of Gram-negative bacteria. Whereas the primary mechanism of Gram-positive bacteria is due to the mutation of the targets, penicillin-binding proteins (PBPs), that of Gram-negative bacteria is caused by the expression of β-lactamases [221,222]. The former mechanism of Gram-positive bacteria is similar to the Gram-positive bacterial β-lactam producers [223]. Among Gram-positive bacteria, *S. pneumoniae*, *Enterococcus faecium*, and *S. aureus* are clinically important, β-lactam targeting pathogenic bacteria. *S. pneumoniae* contains six PBPs for the construction of its peptidoglycan. Three PBPs (PBP1a, PBP1b and PBP2a) are bifunctional class A PBPs, having both transglycosylase and transpeptidase activities, two PBPs (PBP2b and PBP2x) are class B transpeptidases, and the sixth PBP (PBP3) is a class C DD-carboxypeptidase. Resistance to β-lactam antibiotics by *S. pneumoniae* is the consequence of extensive and complementary mosaic mutations of two key β-lactam target enzymes, PBP2b and/or PBP2x, and compensatory PBP1a [224,225]. In addition, single nucleotide polymorphisms are also associated with β-lactam resistance within pneumococcal genes connected with a couple of the pathways including the peptidoglycan synthesis pathway [226]. Interestingly, unlike in many bacterial pathogens the expression of β-lactamases has not been reported in *S. pneumoniae*, although genes for four enzymes (GB Nos. NP_357719, NP_358132, NP359083, and NP_358185) belonging to the metallo-β-lactamase superfamily are detectable in its genome. NP_358185 is an RNase Z/an arylsulfatase, NP_359083 is related to glyoxalase II, while NP_358132 and NP_357719 are distantly related to metallo-β-lactamases from *B. cereus* (GB No.AAA22276 and AAA22562) and *Bacteroides fragilis* (GB No.AAA22904) [227].

Enterococci are resident bacterial flora isolated from the ileum, oral cavity and vulval region and usually asymptomatic. Recently, however, multidrug-resistant *E. faecium* emerged as a major threat to human health. The main mechanism of resistance of multidrug-resistant *E. faecium* to β-lactams is the mutation and overexpression of PBP5 belonging to class B PBPs like *S. pneumoniae* PBP2b and PBP2x and *S. aureus* PBP2a. Mutations I499T, E629V, and the introduction of an additional Ser466 to the PBP5 gene, and the combinations of these mutations increase the minimum inhibitory concentrations (MICs) of β-lactams to different β-lactams [221]. Additionally, d,l-peptidoglycan transpeptidases may be involved in the resistance. They are implicated in the formation of second type of cross-links, the 3→3 cross-links, and by-pass the classical PBP pathway. These enzymes are detected in in vitro selected *E. faecium*, wild type *M. tuberculosis*, *M. abscessus*, and *C. difficile*. They have a cysteine residue as an active site instead of a serine residue in PBPs, and are sensitive to carbapenem group β-lactams, but resistant to ampicillin and cephem group β-lactams [228].

*S. aureus* is a human commensal, Gram-positive bacteria. Two mechanisms confer resistance in *S. aureus*. The first is the production of PC1 β-lactamase encoded by *blaZ*, which inactivates β-lactams by hydrolysis of its β-lactam ring [229]. *blaZ* products are classified into four groups, A, B, C and D on the basis of their serotype and substrate specificity. *blaZ* is mostly located on plasmids. However, the leading role in the resistance has been replaced by PBPs at the present time, although β-lactamases emerged as the initial resistance mechanism in *S. aureus*. Methicillin-resistant *Staphylococcus aureus* (MRSA), which is resistant to virtually all β-lactam antibiotics, has also acquired the *mecA* gene encoding PBP2a. Additional genes such as *fem* and *aux* have been identified to be necessary for methicillin resistance [230]. Clinical isolates of *S. aureus* with methicillin-resistant and high-level ceftaroline resistant phenotype have PBP2a carrying two contiguous substitutions Y446N and E447K in the cephalosporin-binding pocket of the transpeptidase domain [231].

In contrast to Gram-positive bacteria, Gram-negative bacteria have a thin peptidoglycan cell wall sandwiched between their inner and outer membranes. In Gram-negative bacteria, the primary resistance mechanism to β-lactams is the degradation of β-lactams by β-lactamases especially carbapenem-hydrolysing β-lactamases called as carbapenemases. These enzymes possessing a variety of properties belong to Ambler class A, B, and D. The class A carbapenemases consist primarily of the *Klebsiella pneumoniae* carbapenemase (KPC), *Serratia marcescens* enzyme (SME), and imipenem carbapenemase (IMP). The class B carbapenemases are referred to metallo-β-lactamases (MBLs), and include Verona integron-encoded metallo-β-lactamase (VIM), and New Delhi metallo-β-lactamase (NDM). The class D enzymes consist of the OXA β-lactamases such as OXA23, OXA24, OXA48 and OXA58, which are able to hydrolyze carbapenems [232]. In addition, these carbapenemases exist together with at least one other β-lactamase, and are able to hydrolyze substantially all β-lactam antibiotics. Furthermore, most of the genes for the carbapenemases are located on transferrable plasmids flanked by transposable elements, permitting endless transfer and dissemination between bacteria of different species in different environments, sometimes crossing over between Gram-positive and Gram-negative bacteria [233]. The amino acid sequence of Tcur2040 β-lactamase (GB No.WP_012852391) from *Thermomonospora curvata*, a thermophilic Gram-positive Actinobacterial species, is also closely related to that of class D β-lactamases such as OXA5 (GB No.X58272) and OXA27 (GB No.AAC15074, partial) from Gram-negative bacteria [234]. Similarity values are 1.1e-46 and 1.8e-43 at amino acid sequence level, and 1.1e-14 and 2.8e-15 at nucleotide sequence level, respectively. Consequently, bacterial species usually susceptible to carbapenems such as *Enterobacteriaceae*, *Acinetobacter baumannii*, *P. aeruginosa*, and *K. pneumoniae* have acquired the ability to hydrolyze β-lactams and make them highly resistant to most β-lactam antibiotics. Moreover, efflux pumps of the RND superfamily such as AcrB of *E. coli* and MexB of *P. aeruginosa* are reported to play an important role in producing intrinsic multidrug resistance including β-lactam antibiotics in Gram-negative bacteria [235].

In summary, the resistance mechanism in β-lactam producers is primarily due to the intrinsic resistant PBPs. In contrast, the resistance mechanism in Gram-positive pathogenic bacteria is mainly attributed to the mutation of PBPs, while that in Gram-negative bacteria resides in the acquisition of β-lactamases. Intriguingly, the amino acid sequence of PBP2x of *S. pneumoniae* (GB No.AFC91889) is more closely related to those of group VIII-5 *Streptomyces* PBPs such as SCO2090, SCATT_12070, SCLAV_1301, and SSHG_01149 [223] than to those of PBP1a (GB No.AFC91821) and PBP2b (GB No.AAC95433) of *S. pneumoniae*, to PBP5 of *E. faecium* (GB No.AIG13035), and to PBP2a of *S. aureus* (GB No.AVI00630). E values in the former cases are in the order of 1e-20, but those in the latter cases are in the order of 1e-10, suggesting that group VIII-5 PBPs are somehow related to self-resistance in *Streptomyces* species besides group VIII-6 PBPs [223]. On the other hand, the amino acid sequences of class A carbapenemases in Gram-negative bacteria such as SME3, IMI3, NMC-A, KPC1 and GES2 are closely related to those of actinobacterial β-lactamases such as SCAB_38731, SAV_4452, SACE_1374, and Amir_2178 (Appendix A). For example, E values between KPC1 (GB No.AF297554_1) and these sequences are 5.9e-35, 6.1e-34, 2.8e-40 and 4.0e-33 at amino acid sequence level, and 2.0e-28, 7.6e-31, 8.3e-43 and 1.8e-28 at nucleotide sequence level, respectively. However, sequence similarities between class B carbapenemases and actinobacterial β-lactamases are not so high. For example, E values between IMP1 (GB No.AXQ85786) and Tcur_2765 and Ppa_0914 are 2.2e-08 and 3e-07, respectively. In any case, it is quite surprising that the sequences in Gram-positive pathogenic PBPs are considerably homologous to those in actinobacterial PBPs and, in addition, the Gram-negative pathogenic β-lactamases have some similarity to actinobacterial β-lactamases [234].

### 3.2. Glycopeptides, Lipopeptides and Related Antibiotics

The glycopeptide and lipoglycopeptide antibiotics such as vancomycin and teicoplanin show antibacterial activity against Gram-positive bacteria through binding to the D-alanyl-D-alanine terminus of the lipid II bacterial cell wall precursor and sequestrating the lipid II substrate, resulting in the inhibition of peptidoglycan biosynthesis. They are supposed to be the last resort for the treatment of infections caused by methicillin-resistant *S. aureus* (MRSA) and *Enterococcus* species. Recently, however, the resistance to these antibiotics is emerging and conferring a terrible threat to human health [236]. In contrast, Gram-negative bacteria are intrinsically resistant to these antibiotics because the presence of the outer membrane prevents these molecules from reaching the target. They are biosynthesized by non-ribosomal peptide synthetases [237,238]. The biosynthetic gene cluster of vancomycin was cloned from *Amycolatopsis orientalis* [239] (GB Nos. HE589771 and HQ679900). The ABC transporter (GB No, CCD33134) and VanHAX resistance cassette genes are present within and at the end of the cluster. VanH is a dehydrogenase that converts pyruvate into d-lactate, VanA is a d-Ala-d-Lac ligase, and VanX is a d-Ala-d-Ala dipeptidase that cleaves residual d-Ala-d-Ala dipeptide. Moreover, *vanHAX* genes are detectable not only in vancomycin-related glycopeptide producing *Actinoplanes teichomyceticus* [240] and *Streptomyces toyokaensis* [241] but also in non-producing *Streptomyces coelicolor* (GB No. AL939117, SCO3594~SCO3596). The biosynthetic gene clusters for vancomycin-related glycopeptide or lipoglycopeptide antibiotics such as balhimycin, chloroeremomycin, teicoplanin, A47934, complestatin, A40926, and pekiskomycin were also cloned [240,241,242,243,244,245,246,247,248,249,250]. The ABC transporters are present in all these clusters. Intriguingly, however, glycopeptide antibiotic A40926 producer *Nonomuraea* species does not possess the canonical *vanHAX* genes, instead it possesses *vanY* gene encoding a novel d,d-peptidase/d,d-carboxypeptidase involved in the self-resistance and peptidoglycan maturation [248,249]. VanHAX enzymes together with two component regulatory system VanSR are implicated in the resistance and regulatory mechanisms in VRE by redirecting a portion of the peptidoglycan biosynthetic pathway, whereas *vanSR* genes are missing in the DNA flanking the *vanHAX* cluster in balhimycin producer, and *vanHAX* genes are constitutively expressed [251,252]. It is speculated, therefore, that *vanHAX* and their regulatory systems have sophisticatedly evolved depending on the divergent environment surrounding glycopeptide antibiotic producers and related soil-dwelling *Actinobacteria* [237,253].

On the contrary, in the pathogenic bacteria the first VRE were reported in 1988 [254], and now they are ubiquitously distributed in hospitals as well as in the environment throughout the world. The major resistance mechanism is due to two causes: the first is the substitution of the target d-Ala-d-Ala residues in the peptidoglycan by low affinity termini (d-Ala-d-Lac or d-Ala-d-Ser) and the second is the removal of d-Ala-d-Ala precursors [255]. Ten types of glycopeptide resistance determinants have been reported. Among them, *VanA* and *VanB* genotypes predominate worldwide. The *VanA* type resistance element was originally detected on a plasmid in an *E. faecium* clinical isolate, and its prevalence has occurred not only in enterococci but also in MRSA, resulting in almost no therapeutic avenues to treat these pathogenic bacteria now. The second type of resistance is mediated by the cooperative function of *VanX* and *VanY* which is located adjacent to *vanHAX* gene cluster. However, this type of resistance is rare comparing to *VanA* and *VanB* types. It is interesting that the resistance gene organization in enterococci is very similar to that in the glycopeptide producers [256] and, in addition, the amino acid sequences of these proteins in enterococci are extremely similar to those in glycopeptide producers and the related *Actinobacteria*. For example, E values between VanA of *E. faecium* (GB No.APC57471) and that of *A*. *orientalis* (GB No.CCD33129), *A. mediterranei* (GB No.WP_013230018), *A. balhimycina* (GB No.RSM46375), *A. teichomyceticus* (GB No.CAE53344), *S. toyocaensis* (GB No.AAC23582), *Streptomyces* sp. WAC1420 (GB No.AGF91737), *S. coelicolor* A3(2) (GB No.NP_627790), and VanB of *E. faecalis* (GB No.AKJ75209) and VanC of *E. casseliflavus* (GB No.WP_12847813) are 1.3e-105, 6.7e-92, 3.6e-103, 2.8e-90, 7.8e-99, 6.4e-102, 2.6e-101, 5.4e-117, and 1.6e-45 at the amino acid sequence level, respectively. More surprisingly, the sequence similarity at the nucleotide sequence level is exceedingly high: the E values in the above pairs are 1.9e-107, 4e-115, 2.5e-102, 4.9e-127, 5.4e-96, 3.3e-98, 1.2e-116, 9e-200, and 1.8e-24, respectively, although the GC contents between enterococci and *Actinobacteria* are quite different. *E. coli* and other Gram-negative bacteria show intrinsic resistance to vancomycin which results from the permeability barrier imposed by the outer membrane as described above [257].

Moenomycin group antibiotics belong to phosphoglycolipids produced by *S. ghanaensis* and inhibit peptidoglycan glycosyltransferases involved in the penultimate step of bacterial cell wall biosynthesis. Although they have been used as animal growth promoters for a couple of decades, no report has been published on significant resistance to moenomycins. The moenomycin A biosynthetic gene cluster was cloned as two separate clusters: one cluster contains three genes involved in A ring assembly and the other cluster contains the genes involved in the assembly of the phosphoglycolipid pentasaccharide scaffold [258] (GB No.DQ988994). Four transporter genes are present within the cluster. In addition, most actinomycetes including moenomycin non-producers are reported to be intrinsically resistant to moenomycin A [259]. Furthermore, genome mining revealed that presumed moenomycin gene clusters are present in *S. clavuligerus*, and γ-proteobacteria *Photorhabdus luminescens* and *P. asymbiotica*. Teichomycin A_1_ is a phosphoglycolipid antibiotic related to moenomycin, and was isolated from *A. teichomyceticus*, teicoplanin-producer, although its exact chemical structure has not been determined yet. The biosynthetic gene cluster was cloned from *A. teichomyceticus* [260] (GB No.KU726098). Although the overall architectures of teichomycin and moenomycin gene clusters are quite different, out of 18 identified teichomycin genes 16 teichomycin genes are orthologous in the moenomycin gene cluster. Within the cluster four transporter genes are present that may be involved in the self-resistance.

Daptomycin is a cyclic lipodepsipeptide produced by *S. roseosporus* that is clinically used for the treatment of severe infections with Gram-positive bacteria. It was reported that the effect of daptomycin is due to the alteration of the permeabilization and depolarization of the bacterial cell membrane by interacting with Ca^2+^ ion and phosphatidylglycerol in the membrane [261]. However, the detailed mechanism of action has not been understood. The biosynthetic gene clusters were cloned from *S. roseosporus* and *Saccharomonospora viridis* [262,263]. Adjacent to the core biosynthetic region, two transporters and a membrane protein genes (*dptM*, *dptN* and *dptP*; *dptN-sv*, *dptM-sv* and *dptP-sv*), that may be involved in the self-resistance, are present in *S. roseosporus* and *S. viridis* genomes, respectively. Daptomycin non-susceptible *S. aureus* clinical isolates have been obtained very rarely at the present time. They are ascribed to the mutation of the genes involved in cell membrane homeostasis, such as *mprF* encoding lysyl-phosphatidyl glycerol synthetase; *yycG*/*walK* encoding the synthesis of a histidine kinase sensor and a regulator; *rpoB* and *rpoC* encoding RNA polymerase β and β’ subunits respectively, and others [264,265]. Daptomycin resistance associated with host defense cationic peptides, that related to cardiolipin synthase mutation, and that associated with the mutations within the genes encoding the LiaFSR regulatory system and YycFGHIJ were reported in clinical isolates of *S. aureus*, and *enterococci* [266,267,268].

Friulimicin and laspartomycin are cyclic lipopeptide antibiotics isolated from *Actinoplanes friuliensis* and *S. viridochromogenes*, and show antimicrobial activity against broad spectrum of Gram-positive pathogenic bacteria including MRSA and VRE. While friulimicin B inhibits the cell wall precursor cycle, the exact mechanism of action of laspartomycin remains to be clarified [269]. The biosynthetic gene cluster for friulimicin was cloned from *A*. *friuliensis* [270]. Three transporter genes are present within the cluster. The biosynthetic gene cluster for laspartomycin was cloned and compared with that of friulimicin [271]. Three transporter genes are present within the cluster. Surfactin is a cyclic lipopeptide that is biosynthesized by non-ribosome peptide synthetases (NRPS) in *Bacillus* species. The whole genome sequencing of *B. amyloliquefaciens* FZB42 and *B. subtilis* 916 revealed that the genomes contain three NRPS gene clusters encoding surfactin, bacillomycin D, and fengycin, and four NRPS gene clusters encoding surfactin, bacillomycin L, fengycin, and locillomycin, respectively [272,273,274]. Many transporter genes exist in the genomes. At least some of them may be implicated in the self-resistance. Tsuge et al. reported that YerP protein was involved in the self-resistance to surfactin in *B. subtilis* [275]. The amino acid sequence analysis indicates that YerP may correspond to KO64_03555 protein in *B. subtilis* 916, which is assumed to be the swarming motility protein SwrC or the multidrug efflux pump subunit AcrB [274].

### 3.3. Polyene Macrolides

Amphotericin B belongs to a group of polyene macrolide antibiotics and has been used as a medically important antifungal antibiotic for several decades. It exerts the antibiotic activity by disruption of fungal cell wall synthesis through its binding to sterols, primarily ergosterol in fungal cell wall, resulting in the formation of trans-membrane channels and disturbing the barrier function of the membrane. However, it also binds to cholesterol in mammalian cell membranes to a lesser extent and shows many side effects, especially nephrotoxicity [276]. Moreover, it induces oxidative damage in the cells and modulates the immune system [276,277]. The biosynthetic gene cluster was cloned from *S. nodosus* [278]. Two ABC transporter genes (*amphG* and *amphH*) are detected at the end of the cluster. It is speculated that AmphG and AmphH excrete the drug by forming a heterodimer. Amphotericin B was also produced by *Penicillium nalgiovense* Laxa [279]. Nystatin is a polyene macrolide antibiotic produced by *S. noursei* and an important antifungal agent used for the treatment of superficial mycosis. Dos Santos et al. revealed that the mechanism of nystatin action is, in addition to the action on membrane sterols, dependent on the membrane biophysical properties and lipid composition [280]. The biosynthetic gene cluster for nystatin was cloned from *S. noursei* ATCC11455 [281]. It contains six genes including two transporter genes *nysG* and *nysH* within the cluster that may be involved in ATP-dependent nystatin efflux. Candicidin is an aromatic polyene macrolide antibiotic isolated from several Actinomycetes. The biosynthetic gene clusters were cloned [282,283]. They contain two ABC transporter genes at the end of the clusters. Natamycin, also called pimaricin, tennecetin and natacyn, is a polyene macrolide antibiotic and widely used in pharmacotherapy for topical treatment and in the food industry as a natural food preservative. However, it does not exert its antifungal activity by forming pores and permeabilizing the plasma membrane, as other polyene macrolide antibiotics do. Instead, it inhibits a wide range of essential membrane transporters in an ergosterol-dependent manner [284]. The biosynthetic gene clusters were cloned from *S. natalensis* and *S. chattanoogensis* [285,286]. In the former, two genes for ABC transporters (*pimA* and *pimB*) and, in addition, one gene for an efflux pump (*pimH*) are present within the cluster, whereas *pimH* is absent in the cluster from *S. chattanoogensis* and other polyene macrolide gene clusters [287]. Although natamycin does not have any antimicrobial activity, the overproduction of the drug is thought to be harmful to the producer organisms. The efflux pumps may function in such situation.

Some clinical isolates of *Candida glabrata* show the reduced susceptibility to amphotericin B and nystatin, although polyene macrolides in general show a remarkably low rate of antifungal resistance. The isolates have lower ergosterol contents in their membrane than the wild type due to nonsense or missense mutations in the genes which are involved in the biosynthesis of ergosterol. However, ergosterol levels of clinical isolates of *Aspergillus terreus* are almost identical. On the other hand, there are several reports demonstrating that resistant mutants with similar ergosterol contents convert cell wall composition significantly, especially β-1,3-glucan contents. The other mechanisms such as adaptation to oxidative stress and a decrease in mitochondrial activity are also implicated in the resistance of pathogenic fungi [288,289].

### 3.4. Lantibiotics and Cyclic Peptides

Lantibiotics are members of bacteriocins and belong to a group of ribosomally synthesized, post-translationally modified peptides containing unusual amino acids including the thioether amino acids such as lanthionine. They are produced by certain Gram-positive bacteria, and in general have a desirable feature of low likelihood of promoting the development of bacterial resistance and low cytotoxicity at antimicrobial concentration [290,291,292]. The first lantibiotic, nisin, was discovered in the 1920s and has been used as a food preservative for several decades. Lantibiotics are classified into type A and type B peptides based on their chemical structures and functional properties. Type A lantibiotics are elongated, cationic peptides up to 34 residues in length, whereas type B lantibiotics are globular peptides up to 19 residues in length. Type A lantibiotic nisin binds to and forms a complex with lipid II, a precursor of cell wall peptidoglycan, resulting in the inhibition of peptidoglycan synthesis. Moreover, nisin increases the membrane permeability to lead to the pore formation and the release of ions and small molecules from the target bacteria [293]. Nisin is shown to have antimicrobial activity against both Gram-positive and some Gram-negative pathogens. In addition, nisin is suggested to influence multisets of host immune cells [294]. Type A lantibiotics such as nisin, subtilin, and sublancin, also inhibit the germination of spores from *Bacillus* and *Clostridium* species. Type B lantibiotic mersacidin inhibits cell wall biosynthesis by forming a complex with lipid II but does not form pores in the cytoplasmic membrane [295]. The biosynthetic gene cluster for nisin was cloned [296] (GB No.HM219853). The eleven genes required for nisin biosynthesis are located in the gene cluster on Tn7276. *nisFEG* located at the end of the cluster encode putative transporter proteins and *nisI* within the cluster encodes a nisin-binding lipoprotein that is involved in immunity. The gene cluster for type AI lantibiotic subtilin was cloned from *Bacillus subtilis* subsp. *spizizenii* ATCC 6633 [297]. Compared to nisin, the subtilin gene cluster lacks the gene corresponding to *nisP*, which codes for an extracellular subtilisin-like protease involved in precursor processing. Four genes *spaIFEG* located within the cluster are implicated in the immunity. *spaT* encoding a transporter is also within the cluster. Type AI lantibiotic microbisporicin produced by actinomycetes *Microbispora corallina* contains chlorinated tryptophan and dihydroproline residues. It shows antimicrobial activity against a wide range of Gram-positive pathogenic bacteria, and inhibits peptidoglycan biosynthesis by binding to lipid II, and some activity against Gram-negative bacteria such as *E. coli*, *Moraxella catarrhalis*, and *H. influenza*, although the exact mechanism remains to be clarified. The biosynthetic gene cluster was cloned and characterized [298] (GB No.HM536998). Four transporter/immunity genes (*mibT*, *mibU*, *mib*E, *mibF*) are identified within the cluster. Type AII lantibiotic lacticin 481 forms a complex with lipid II and inhibits PBP1b-catalyzed peptidoglycan biosynthesis [299]. The biosynthetic gene cluster for lacticin 481 was cloned and characterized [300]. Four genes *lctTFEG* within the cluster are involved in drug transporter or immunity. Among them, at least three genes *lctFEG* are necessary to provide immunity to lacticin 481. The gene corresponding to *nisI* is missing in the cluster. The type AII bacteriocin sublancin 168 differs from other lantibiotics of this group due to the presence of two disulfide linkages [301]. The mechanism of action of sublancin is also different from other lantibiotics, that is, that sublancin negatively affects DNA replication, transcription and translation without affecting cell wall biosynthesis [302]. The gene cluster was cloned from *B. subtilis* 168 [301]. The ABC transporter gene (*sunT*) and immunity protein gene (*sunI*) are located within the cluster. The phosphoenolpyruvate: sugar phosphotransferase system is involved in the sensitivity to the glucosylated sublancin [303]. Cinnamycin isolated from *S. cinnamoneus* belongs to the type B lantibiotics, and binds to phosphatidylethanolamine in the inner layer of the plasma membrane, inducing membrane reorganization such as membrane fusion and the alteration of morphology of the membrane [304]. The gene cluster was cloned and characterized [305]. Two component transporter genes *cinT* and *cinH* are located in the middle of the cluster. Another actinomycete lantibiotic actagardine is a tetracyclic, type B lantibiotic. It was reported that actagardine inhibits the peptidoglycan transglycosylase, but the details remain to be elucidated [306]. The biosynthetic gene cluster for actagardine was cloned and characterized [307]. Two transporter genes, *garH* and *garT*, are present within the cluster and three genes, *orf16*, *orf17*, and *orf18*, are present adjacent to the cluster. The bacteriocin enterocin F4-9 is not a lantibiotic but an O-linked glycopeptide isolated from *Enterococcus faecalis* [308]. However, the organization of gene cluster is analogous to that of sublancin [301].

Although nisin has been used for decades as a natural preservative in the food industry, there are only few cases of naturally occurring lantibiotic resistance. However, human pathogenic bacteria such as *S. aureus* and *Streptococcus agalactiae* confer inherent resistance against lantibiotics such as nisin. There are several mechanisms for the intrinsic resistance: First, nisin resistance protein (NSR), a membrane-associated protease, is detected in *S. lactis*, *S. aureus*, *S. agalactiae* and other pathogenic bacteria [309,310]. The genes for these proteins are regulated by specific two-component systems. Second, ABC transporter is involved in the drug excretion from inside the cells. The expression of the ABC transporters is also regulated by two-component systems [291,311]. Third is the modifications of cell wall and membrane. The D-alanylation of teichoic acids alters the net charge distribution in cell wall, making cationic lantibiotics be repelled from the cell envelope of target bacteria such as *S. aureus*, *S. pneumoniae*, *Clostridium difficile*, and *B. cereus* [312]. The overexpression of the penicillin-binding proteins makes *Listeria monocytogenes* and *S. aureus* resistant to nisin and cell wall-targeting compounds [313]. The changes of the lipid composition and lysine esterification of the hydroxyl groups of phosphatidylglycerol in cell membranes were also reported to be implicated in the resistance of pathogens against lantibiotics [314]. Summarizing these results, the mechanisms of action of lantibiotics in pathogenic bacteria are little different from those in producing bacteria. In producing bacteria the main players are ABC transporters and immunity protein, while in pathogenic bacteria the nisin resistance protein and the modification of cell wall are additional players.

The polymyxins are the cyclic lipopeptide antibiotic family and were first isolated in 1947 from *Bacillus polymyxa* (*Paenibacillus polymyxa*) as antibiotics specifically active against Gram-negative bacteria such as the majority of Enterobacteriaceae as well as *Acinetobacter baumannii* and *P. aeruginosa*. Although their clinical use were once gradually decreased by the nephrotoxicity and neurological effects, they have resurged as the last-line drugs against extremely resistant Gram-negative bacteria, due to the global dissemination of these bacteria. The positively charged polymyxins bind to the negatively charged lipid A part of lipopolysaccharide (LPS), exclusively present in Gram-negative bacteria and located in the outer membrane [315]. The polymyxin biosynthetic gene cluster was cloned from *P. polymyxa* and characterized [316]. Two transporter genes *pmxC* and *pmxD* are present in the middle of the cluster. A number of Gram-negative bacteria such as *Brucella* spp., *Edwardsiella* spp., and *Serratia* spp. are naturally resistant to polymyxins. The mechanism is hypothesized to be due to the addition of cationic molecules to LPS. Recently, the acquired resistant strains to polymyxins are increasing in clinical isolates. The mechanisms of the resistance are various modifications of LPS and the use of efflux pumps, some of which are encoded by genes located on plasmids [317,318]. Bacitracin is a cyclic peptide antibiotic produced by several *B. licheniformis* strains. It is synthesized non-ribosomally by the large multienzyme complex BacABC. It is used as a single ointment or in combination as a triple therapy ointment with neomycin and polymyxin B, and is active against Gram-positive bacteria including *Staphylococcus*, *Streptococcus*, *Corynebacterium*, and *Clostridium* species. Bacitracin interferes with bacterial cell-wall synthesis by binding undecaprenyl pyrophosphate, a lipid carrier that serves as a critical intermediate in cell wall biosynthesis [319]. The biosynthetic gene cluster of bacitracin was cloned and characterized [320,321]. The transporter encoding genes *bcrABC* are located about 3kb downstream of the bacitracin biosynthetic operon *bacABC*. The two-component regulatory system BacRS is present between these clusters and is implicated in the regulation of the self-resistance of *B. licheniformis*.

### 3.5. Other Cell Wall/Membrane Synthesis Inhibitors

Capuramycin-type nucleoside antibiotics such as A-500359s and A-102395 show antimicrobial activity against Gram-positive and Gram-negative bacteria and, in addition, *M. tuberculosis*. Recently, this type of antibiotics gains a considerable attention as potent anti-tuberculosis drugs, because multidrug resistant *M. tuberculosis* strains pose significant threats to human health [322]. These antibiotics inhibit bacterial translocase I, a ubiquitous and essential enzyme that acts in peptidoglycan cell wall biosynthesis. The biosynthetic gene clusters for A-500359s and A-102395 were cloned from *S. griseus* and *Amycolatopsis* sp., respectively [323,324]. The former contains genes encoding aminoglycoside phosphotransferase (*orf21*) and four transporters (*orf19*, *orf20*, *orf30* and *orf35*), and the latter contains genes for phosphotransferase (*cpr17*), but three transporter genes are replaced by three genes for transposases. Instead, four transporter genes (*orf58*, *orf59*, *orf60* and *orf61*) exit near the end of the cluster [324]. The phosphotransferase (ORF21) phosphorylates the unsaturated hexuronic acid, a component of A-500359s. These genes are supposed to be implicated in the self-resistance. The biosynthetic gene cluster for A-503083s was also cloned from *Streptomyces* sp. SANK 62, 799 [325].

d-Cycloserine is a natural metabolite of *S. lavendulae* and *S. garyphalus* and is a structural analog of d-alanine. It inhibits two sequential enzymes in the cell wall peptidoglycan biosynthetic pathway, alanine racemase and d-Ala-d-Ala ligase [326]. It is used clinically for the treatment of tuberculosis as a second-line drug. The biosynthetic gene cluster for d-cycloserine was cloned [327]. d-Ala-d-Ala ligase and a putative membrane protein (transporter) are proposed to be involved in the self-resistance [327]. Fosfomycin is a revival antibiotic and is a phosphoenolpyruvate analogue produced by *Streptomyces* species. It is active against both Gram-positive and Gram-negative drug-resistant pathogens. Fosfomycin interferes with and inhibits the first step enzyme, UDP-N-acetylglucosamine enolpyruvyl transferase (MurA), of bacterial peptidoglycan biosynthesis [328]. The biosynthetic gene cluster for fosfomycin was cloned from *S. fradiae* and characterized [329,330]. *fomA* and *fomB* within the cluster are proposed to be involved in the self-resistance in *S. wedmorensis* [329]. Mutations of *murA*, a salvage pathway in peptidoglycan biosynthesis, mutations in the uptake systems, and fosfomycin-modifying enzymes are described to function in the fosfomycin resistance in pathogenic bacteria [331]. In *M. tuberculosis*, mutations of alanine racemase, L-alanine dehydrogenase are reported to be implicated in the resistance [332,333]. On the other hand, fosfomycin resistance is very rare in pathogenic bacteria [334].

## 4. DNA Synthesis Inhibitors and Related Antibiotics

### 4.1. Bleomycin and Related Antitumor Antibiotics

Bleomycins are glycopeptide-derived antibiotics produced non-ribosomally by *S. verticillus* and are clinically valuable natural products used for the treatment of several types of tumors [335]. Bleomycins exert their biological activity though metal-dependent oxidative cleavage of DNA and RNA, but are not cell wall inhibitors such as vancomycin and teicoplanin. The biosynthetic gene cluster for bleomycin was cloned from *S. verticillus* ATCC15003 as 77kb DNA fragment [336] (GB No.AF210249). The three self-resistance-related genes, *blmA*, *blmB* and *orf7* encoding the bleomycin-binding protein, the bleomycin acetyltransferase, and ABC transporter respectively, are present at the end of the cluster [337,338,339] (GB No. L26955). *orf29*, which is located at another end of the cluster, may also be involved in the self-resistance by transporting the drug [336]. Tallysomycin and zorbamycin are members of bleomycin family antitumor antibiotics. Their biosynthetic gene clusters were cloned from *Streptoalloteichus hindustanus* and *S. flavoviridis*, respectively [340,341]. The gene cluster for tallysomycin contains one each of the genes for ABC transporter (*tlmT*), the tallysomycin-binding protein (*tlmA*) and an N-acetyltransferase (*tlmB*), whereas that for zorbamycin contains three genes for ABC transporters (*orf36*, *orf37* and *orf38*) and the zorbamycin-binding protein (*zbmA*) within the clusters but lacks the zorbamycin acetyltransferase gene. Interestingly, BlmB and TlmB acetylate the metal-free forms of four bleomycin family members, that is, bleomycin, phleomycin, tallysomycin and zorbamycin, and BlmB can provide resistance to zorbamycin in *S. flavoviridis*, a zorbamycin-producer [342]. Bleomycin can be metabolically inactivated in normal and tumor cells by bleomycin hydolases. These hydrolases are widely distributed in mammals, yeast, and bacteria [343]. DNA strand breaks caused by oxidative damage can be repaired by polynucleotide kinase 3′-phosphatase, the *pnkp* gene product [344]. PNKP and its homologs are detectable in mammalian cells, yeast as well as bacteria. In addition, the bleomycin-binding proteins are found in MRSA and some pathogenic bacteria [345]. These three mechanisms may function as the major role in the bleomycin resistance in the clinical field.

### 4.2. Quinone and Related Antitumor and Antimicrobial Antibiotics

The angucycline group antibiotics are referred to the characteristic four-ring frame of the aglycone moiety, including tetracycline group antibiotics and anthracycline group antibiotics [346,347]. Daunorubicin, aclacinomycin, and nogalamycin belong to the anthracycline group antitumor antibiotics. Their chemotherapy maintains a prominent role in the treatment of many forms of tumors, as they are listed among the World Health Organization (WHO) model list of essential medicines [348]. However, their clinical use is limited because of the cardiotoxic side effect. The basic mechanisms of cardiotoxicity involve direct pathways for reactive oxygen species generation and the inhibition of topoisomerase 2B [349]. On the other hand, as anthracyclines mediate diverse molecular effects, the mechanism of their cytotoxicity includes multiple pathways: intercalation into DNA, generation of free radicals, DNA binding and alkylation, inhibition of helicase, inhibition of topoisomerases, and so on [350]. The doxorubicin (14-hydroxydaunorubicin, adriamycin) and daunorubicin biosynthetic gene cluster were cloned from *S. peucetius* and characterized [351,352]. Interestingly, *S. peucetius* ATCC 29, 050 produces daunorubicin but not doxorubicin, whereas *S. peucetius* ATCC 29, 052 produces both daunorubicin and doxorubicin. DrrA and DrrB form a transporter of the antibiotic [353], and DrrC is a UvrS-like DNA repair protein [354]. These are involved in the self-resistance. Daunorubicin forms a specific complex with a secreted serine protease and the protease sequesters daunorubicin to prevent its entry into the cells [355]. DrrD was reported to be implicated in the resistance. However, the detailed mechanism has not been analyzed so far. Aclacinomycin A (aclarubicin), a trisaccharide anthracycline, is shown to be active in patients with acute myeloblastic leukemia, but induces late cardiac toxicity. Aclacinomycin A inhibits topoisomerase I and topoisomerase II. The biosynthetic gene cluster for aclacinomycin was cloned from *S. galilaeus* [356,357]. AcrV (ABC transporter ATP binding component) and AcrW (ABC transporter transmembrane protein) are present at the end of the cluster. Nogalamycin is an anthracycline group antibiotic produced by *S. nogalater*. The chemical structure is unique in that one of the carbohydrate units of the molecule, nogalamine, is attached both via a carbon-carbon bond and a classical O-glycosidic linkage to the aglycone. It is an antitumor antibiotic and belongs to a threading intercalator, in contrast to daunorubicin which belongs to a non-threading classical intercalators. The biosynthetic gene cluster for nogalamycin was cloned from *S. nogalater* as a 20kb DNA fragment [358]. The genes encoding SnoO (a nuclear transport factor 2 superfamily member, a polyketide cyclase?) and SnorO (an UvrA-like excinuclease) are identified within the cluster. These may be involved in the self-resistance.

Landomycin, urdamycin and jadomycin belong to angucycline group antibiotics. Landomycin E is produced by *S. globisporus*, and shows antitumor activity which is only mildly affected by multidrug resistance-mediated drug efflux. It is reported that landomycin E arrests tumor cell cycle progression, and induces apoptosis by activation of initiator procaspase-16 [359]. The biosynthetic gene cluster for landomycin E was cloned from *S. globisporus* [360], and that for landomycin A was cloned from *S. cyanogenus* [361]. Three genes *lndJ*, *lndW*, and *lndW2* encoding a transporter, an ABC transporter, and an ATPase of ABC transporter were detected within and at the end of the cluster, respectively. Urdamycin is an angucycline-type antibiotic and biologically active against Gram-positive bacteria and stem cells of murine L1210 leukemia. The biosynthetic gene cluster was cloned from *S. fradiae* [362]. Two transporter genes, *urdJ* and *urdJ2*, are detected within the cluster. Aquayamycin and rabelomycin have similar chemical skeleton as urdamycins. Jadomycins are angucycline-type antibiotics and inhibit topoisomerases and promote apoptosis [363]. The biosynthetic gene cluster for jadomycin was cloned from *S. venezuelae* [364]. A MFS type transporter gene *jadL* is located within the cluster.

Actinorhodin, granaticin, and medermycin are the benzoisochromanequinone-type antibiotics. Actinorhodin is a blue-pigmented, pH-responsive antibiotic. It shows a weak antibacterial activity against Gram-positive bacteria and causes oxidative damage of multiple cellular targets including DNA, proteins and cell envelope [365]. The biosynthetic gene cluster for actinorhodin was cloned from *S. coelicolor* and characterized [366,367]. Three transporter genes *actII-orf2*, *actII-orf3* and *actVA-orf1*, corresponding to SCO5083, SCO5084 and SCO5076, respectively, are detected within the cluster. Granaticin is an antibiotic produced by *Streptomyces* species and shows an antibacterial activity against Gram-positive bacteria and antitumor activity against P-388 lymphocytic leukemia in mice and human oral epidermoid carcinoma cells [368]. It inhibits rRNA maturation and cell cycle specificity. The biosynthetic gene cluster was cloned from *S. violaceoruber* [369,370]. One transporter gene *orf15*, which is corresponding to *actII-orf2* in *S. coelicolor*, is detectable within the cluster. Medermycin is an antibiotic active against Gram-positive bacteria. It inhibits platelet aggregation. The biosynthetic gene cluster was cloned from *Streptomyces* sp. AM-7161 [371]. One transporter gene *orf25* is detected within the cluster. Mitomycins were isolated from *S. lavendulae* as antitumor antibiotics more than 60 years ago. They are comprised of aziridine, quinone, and carbamate moieties arranged in a compact pyrroloindole structures. They cross-link DNA with high efficiency and specificity for the sequence CG, and function as alkylating agents only after reductive conversion to highly reactive quinone methides [372]. The biosynthetic gene cluster for mitomycin C was cloned from *S. lavendulae* [373]. Three genes *mcrA*, *mrd*, and *mct* encoding a flavoprotein oxidoreductase, mitomycin-binding protein, and translocase/transporter, respectively, were reported to be involved in the self-resistance [374,375,376]. Two of them are located within the cluster, but *mcrA* is outside of the cluster. Interestingly, two homologous genes to *mcrA*, *mitR* and *mmcM*, are present within the cluster. Yatakemycin is an extraordinarily toxic and DNA alkylating agent with potent antimicrobial and antitumor activity. Together with CC-1065 and duocarmycins, it forms a cyclopropapyrroloindole antibiotic group. The biosynthetic gene cluster for yatakemycin was cloned from *Streptomyces* sp. TP-A2060 [377,378]. Two transporters and one DNA repair enzyme (YtkR6, DNA glycosylase) were reported to be involved in the self-resistance.

Rebeccamycin and staurosporine are the indolocarbazole type antibiotics produced by *Actinobacteria*. Rebeccamycin, a halogen-containing natural product, shows antibacterial and antitumor activities. It inhibits DNA topoisomerase I, suppresses myosin light chain kinase production and induces claudin-5 expression [379]. The biosynthetic gene cluster was cloned from *Saccharothrix* (*Lechevalieria*) *aerocolonigenes* [380,381]. RebT and RebU were reported to be involved in the self-resistance/secretion. The indolocarbazole staurosporine is a potent inhibitor of a variety of protein kinases such as protein kinase C and cyclin-dependent protein kinase. It induces cell cycle arrest, apoptosis, and activation of caspase-3. The biosynthetic gene clusters were cloned from *Streptomyces* sp. TP-A0274 [382] and *S. clavuligerus* ATCC27064. No gene related to the resistance/transporter has been detectable within and adjacent to the gene cluster in both *Streptomyces* sp. TP-A0274 and *S. clavuligerus* ATCC27064. Moreover, no gene related to the resistance/transporter is present within and adjacent to the gene cluster (DDQ41_30655~DDQ41_30725 in GB No. CP029254) in *S. spongiicola* HNM0071, that produces staurosporine. The fact that no gene related to the self-resistance is detectable in these species suggests that these genes were deleted or missing at the beginning from the genome due to the unnecessity for their existence because of the lack of antimicrobial activity. However, details remain to be clarified. Indolmycin is an antibiotic produced by *S. griseus* and *Pseudoalteromonas luteoviolacea*, a marine γ-proteobacterium [383,384], and shows an antimicrobial activity against human pathogens, including MRSA and *H. pylori*. It competes with tryptophan for binding to tryptophanyl-tRNA synthetase as a natural tryptophan analog. The biosynthetic gene cluster for indolmycin was cloned from *S. griseus* ATCC12648 [385]. However, no resistance-related gene is detectable. Interestingly, *S. griseus* NBRS13350 has two genes encoding tryptophanyl-tRNA synthetases (SGR_2702 and SGR_3809). Among them, the SGR_3809 gene encodes an indolmycin-resistant enzyme. Moreover, a cDNA 93% identical in sequence to SGR_3809 was detected in *S. griseus* ATCC12648, the indolmycin producer, indicating that an auxiliary tryptophanyl-tRNA synthetase has a role in the self-resistance [386].

Chromomycin A_3_ and mithramycin are aureolic acid-type antitumor antibiotics. Chromomycin A_3_ forms dimeric complexes with divalent cations and binds to the GC rich sequence of DNA to inhibit DNA replication and transcription [387]. It shows antimicrobial activity against Gram-positive bacteria and inhibits the growth of several lines of tumor cells. The biosynthetic gene cluster was cloned from *S. griseus* [388]. It contains three genes at the end of the cluster that are involved in the self-resistance. The *cmrA* and *cmrB* genes encode the ABC transporters, and *cmrX* encodes an UvrA-like UV repair nuclease. The CmrAB ABC transporter confers a high level resistance to the biosynthetic intermediate [389]. The aureolic acid group compound mithramycin is an antitumor antibiotic produced by *S. argillaceus*. The biosynthetic genes for mithramycin were cloned from *S. argillaceus* [390,391]. Three genes *mtrA*, *mtrB*, and *mtrX*, supposed to be involved in the self-resistance are located at the end of the cluster. *mtrA* and *mtrB* encode a transporter and *mtrX* encodes the UV-repair system.

Thiocoraline, echinomycin and triostin are thiodepsipeptide antibiotics produced by several species of actinomycetes. This group of compounds belongs to a large family of bisintercalators based on their nonribosomally biosynthesized peptide cores [392]. These compounds intercalate with high affinity into the minor groove of DNA and show their potent activity on various tumors, viruses and bacteria. Thiocoraline is a member of the two-fold symmetric bicyclic bisintercalators produced by two marine *Micromonospora* species. This compound shows antibacterial activity against Gram-positive bacteria and antitumor activity against various human cancer cell lines. It inhibits DNA polymerase α and promotes cell cycle arrest. The biosynthetic gene cluster was cloned from *Micromonospora* sp. ML1 as 65kb DNA fragment [393]. Within the cluster, there are two transporter genes (*tioC* and *tioD*), uvrA-like gene (*tioU*) and *tioX* encoding a protein that could be involved in the sequestration of thiocoraline [393,394]. Echinomycin/quinomycin A is a pseudosymmetric bicyclic bisintercalator produced by several *Streptomyces* species. This compound was reported to specifically inhibit binding of hypoxia-inducible factor-1 (HIF-1) to the hypoxia-responsive element (HRE) sequence contained in the vascular endotherial growth factor (VEGF) promoter. The biosynthetic gene clusters were cloned from *S. lasaliensis* [395], and *S. griseovariabilis* [396]. While the cluster in *S. lasaliensi* contains *uvrA*-like gene (*ecm16*), but not two transporter genes, that in *S. griseovariabilis* contains uvrA-like gene (*qui10*) as well as two transporter genes (*qui1* and *qui2*). Recent whole genome sequencing of *S. spongiicola* HNM0071 (GB No. CP029254) indicates that six ABC transporters within the cluster might regulate the biosynthesis of echinomycin [397]. Triostin A is the direct precursor of echinomycin, and the *ecm18* gene product *S*-adenosyl-L-methionine (SAM)-dependent methyltransferase is thought to catalyze this transformation [395]. This compound prefers to bind AT-rich DNA sequences over the GC-rich DNA sequences. The biosynthetic gene cluster for triostin A was cloned from *S. triostinicus* [398]. Two transporter genes (*trsD* and *trsE*) and one DNA repair gene (*trsM*) are present within the cluster.

Azinomycin and ficellomycin are hybrid polyketide/nonribosomal peptide natural products possessing an azabicyclohexane ring system, and show antitumor activity by interacting covalently with duplex DNA in the major groove and inducing interstrand crosslinking. The biosynthetic gene cluster for azinomycin B was cloned from *S. sahachiroi* [399]. One transporter gene (*aziE*) is detected within the cluster. The DNA glycosylase (AlkZ), which functions in the reduction and repair of azinomycin B induced DNA damage, and the azinomycin-binding protein (AziR) are involved in the self-resistance in *S. sahachiroi* [400,401]. Ficellomycin exhibits potent in vitro activity against *S. aureus* including MRSA. It was reported that ficellomycin selectively impairs semiconservative DNA replication in DNA polymerase I-deficient *E. coli* [402]. The biosynthetic gene cluster for ficellomycin was cloned from *S. ficellus* [403]. It was proposed that Fic11, Fic12 and Fic42 function as the ABC transporters and Fic14 and Fic45 are MFS transporters.

Novobiocin, coumermycin A_1_ and clorobiocin constitute the aminocoumarin antibiotics. These compounds are potent inhibitors of gyrase, bind to the B subunit of bacterial DNA gyrase, and inhibit the ATP-dependent DNA supercoiling catalyzed by gyrase. DNA gyrase is a type II topoisomerase. The fluoroquinolones are good examples of very successful gyrase-targeted drugs, although they are not antibiotics. The clinical use of aminocoumarin antibiotics are limited due to their poor pharmacological properties, limited solubility in water, and moderate toxicity to human. The biosynthetic gene cluster for novobiocin was cloned from *S. sphaeroids* NCIB11891 as 25.6kb DNA fragment [404]. The cluster contains transporter gene (*novA*) and the novobiocin-resistant gyrase gene (*gyrB^R^*). The novobiocin-producing *S. sphaeroides* possesses two *gyrB* genes, *gyrB^S^* and *gyrB^R^* [405], encoding novobiocin-sensitive predominant form of the enzyme and novobicin-resistant form, respectively. Coumermycin A_1_ has the most complex chemical structure among the three aminocoumarins described above. The coumermycin A_1_ biosynthetic gene cluster was cloned from *S. rishiriensis* DSM40489 [406]. The cluster contains two aminocoumarin-resistant topoisomerase genes, *gyrB^R^* and *parY^R^*. The *parY^R^* gene, located immediately downstream of *gyB^R^* gene, encodes an aminocoumarin-resistant topoisomerase IV subunit. Gyrase as well as topoisomerase IV belong to type II topoisomerases. Clorobiocin is a halogen-containing aminocoumarin antibiotic produced by a various *Streptomyces* species. The clorobiocin biosynthetic gene cluster was cloned from *S. roseochromogenes* DS 12.976 [407]. No transporter gene was detected in the cloned cluster. Interestingly, the clorobiocin gene cluster possesses two aminocoumarin-resistant topoisomerase genes, *gyrB^R^* and *parY^R^* similar to that of coumermycin A_1_. However, *parY^R^* is missing in the cluster of novobiocin [408].

### 4.3. Enediyne Antitumor Antibiotics

The enediyne antitumor antibiotics are a growing family of natural products with novel molecular architecture with triple bonds, and unique biological activity. The clinical usage of the natural compounds of this family are limited mainly because of their toxicity. However, the improvement of the delivery systems and antibody-drug conjugates have paved the way to great clinical success in antitumor chemotherapy [409]. These compounds are classified roughly into two categories. The first category is comprised of the chromoprotein enediyne, possessing a novel nine-membered enediyne chromophore core with a specific associated protein for stabilization. The second category is constituted of the ten-membered ring system and lacks any additional stabilization factor. The nine-membered as well as the ten-membered enediyne compounds bind DNA with high affinity and induce the oxidative DNA strand cleavage. Neocarzinostatin, C-1027, macromomycin, kedarcidin, maduropeptin, and actinoxanthin belong to the first category, whereas calicheamicin and dynemicin belong to the second category. Cyanosporasides are supposed to be degradation products of enediyne compounds [410]. Neocarzinostatin, produced by *S. carzinostaticus*, is composed of a protein moiety and an enediyne chromophore in the molar ratio of 1:1. The amino acid sequence of the protein moiety was determined [411]. The biosynthetic gene cluster for the enediyne chromophore was cloned from *S. carzinostaticus* as 92kb DNA fragment [412]. There are two self-resistance-related genes, *ncsA* encoding apoprotein for the sequestration of the enediyne chromophore, and *ncsA1* encoding efflux pump transporter within the cluster [412]. In addition, mycothiol-dependent detoxication may be involved in the self-resistance [413]. C-1027 is a chromoprotein antitumor antibiotic produced by *S. globisporus* and composed of an apoprotein and the enediyne chromophore. The amino acid sequence of the apoprotein was determined [414]. The biosynthetic gene cluster was cloned from *S. globisporus* [415]. Four genes are implicated in the self-resistance; *cagA*, *sgcB*, *sgcB4*, and *sgcB2*/*orf (-1)* encoding the apoprotein, a transporter, a transporter, and the UvrA-like protein, respectively. Macromomycin/auromomycin isolated from *S. macromomyceticus* is supposed to contain an enediyne chromophore, but the complete structure of the chromophore has not been elucidated yet. The amino acid sequence of the apoprotein (McmA) was determined, which is essential for the maintenance of the stability of the chromophore [416,417]. Kedarcidin with antitumor activity was isolated as a noncovalent complex consisting of an apoprotein and a reactive enediyne chromophore from *Streptoalloteichus* species. The amino acid sequence of the apoprotein was determined [418]. The biosynthetic gene cluster for the kedarcidin chromophore was cloned [419]. The genes for apoprotein (*kedA*) and two transporters (*kedX2* and *kedX*) are present within the cluster. Maduropeptin was isolated from *Actinomadura madurea* ATCC 39144, consisting of an enediyne chromophore embedded in a highly acidic protein. The cloned biosynthetic gene cluster contains genes for the apoprotein (*mdpA*), a transporter (*mdpR3*), and the DNA repair protein (*mdpR4*) [420]. The antitumor protein actinoxanthin was isolated from *Actinomyces globisporus*. The amino acid sequence of the apoprotein (AxnA) was determined [421,422]. The genes for actinoxanthin apoprotein (DIJ69_34170), transporters, and the UvrA-like protein are located on *S. globisporus* TFH56 plasmid pTFSG1 (GB No. CP029362).

Calicheamicin and dynemicin are ten-membered enediyne compounds. A semisynthetic derivative of calicheamicin covalently coupled to a humanized monoclonal antibody specific for the antigen CD33 was approved by FDA as the first antibody-targeted cytotoxic antitumor drug (Mylotarg) for clinical use [409]. The biosynthetic gene cluster was cloned from *Micromonospora echinospora* [423]. The cluster contains genes for putative transporters (*calT1*~*calT7*) and the self-sacrifice proteins (CalC, CalU16, and CalU19) [424]. The self-sacrifice proteins form the complex with calicheamicin, induce the proteolysis of the complex, and inactivate both the highly reactive calicheamicin and the self-sacrifice protein. Dynemicin is a hybrid compound constituted of anthraquinone and enediyne core, which contribute to binding and cleavage of DNA, respectively. The dynemicin biosynthetic gene cluster was cloned from *Micromonospora chersina* [425]. The cluster contains genes for transporters (DynU6, DynT8, and DynT10), dioxygenase (DynE11) which is similar to bleomycin-resistance protein, and a self-sacrifice protein (DynU16). Tiancimycin is a hybrid compound containing anthraquinone and enediyne core. The tiancimycin biosynthetic gene cluster was cloned [426]. Two transporters (TnmT1, TnmT2), one self-sacrifice protein (TnmB), and three dioxygenase/bleomycin-resistance-like proteins (TnmS1, TnmS2, and TnmS3) are detectable within the cluster. The dioxygenase/bleomycin-resistance-like proteins may function in the sequestration of tiancimycin in the producer [427]. Cyanosporasides were isolated from marine actinomycetes genus *Salinispora*. These compounds are speculated to be spontaneous enediyne degradation products [428]. The cyanosporacide biosynthetic gene clusters were cloned from *Salinispora pacifica* CNS-143 and *Streptomyces* sp. CNT-179 [429]. There are three transporter genes (*orf (-6)*, *orf (-5)*, and *cyaR3*) and resistance gene (*cyaR2*) in *S. pacifica*, and four transporter genes (*orf (-5)*, *orf (-3)*, *cynR4* and *orfR5*) in *Streptomyces* sp. CNT-179, respectively, within or adjacent to the clusters.

## 5. Other Antibiotics

The polyketide tetronate compounds bear a tetronate moiety within the structure. These compounds are divided into two classes: tetronates and spirotetronates, and show a vast structural and functional diversity. Some compounds show antimicrobial activity against Gram-positive bacteria, whereas some show antitumor and anti-retroviral activity [430,431]. Tetronomycin is a linear polyether type tetronate inhibiting the growth of Gram-positive bacteria. The tetronomycin biosynthetic gene cluster was cloned from *Streptomyces* sp. NRRL 11, 266 [432]. No self-resistance-related gene is detectable within the cluster. Maklamicin is a small-sized spirotetronate class antibiotic possessing antimicrobial activity against Gram-positive bacteria and moderate cytotoxicity against some human tumor cell lines. The maklamicin biosynthetic gene cluster was cloned from *Micromonospora* sp. NBRC 110, 955 [433]. Three transporter genes are detected within the cluster. Abyssomicins belong to the small-sized spirotetronate class of antibiotics and inhibit the chorismate pathway and folate biosynthesis [434]. Abyssomicin analogs A88696C, D, and F are gastric ATPase inhibitors. The abyssomicin biosynthetic gene cluster was cloned from deep-sea-derived S. *koyangensis* SCSIO 5802 [435]. Five transporter genes (*abmF1*, *abmF2*, *abmF3*, *abmF4*, *abmD*) are present within the cluster. *abmF1*, *abmF2*, *abmF3*, and *abmF4* constitute a four-component ABC transporter-based import system. Chlorothricin is a medium-sized spirotetronate class antibiotic, and inhibits pyruvate carboxylases from rat liver and *Azotobacter vinelandii* and cholesterol biosynthesis. The chlorothricin biosynthetic gene cluster was cloned from *S. antibioticus* DSM 40, 725 [436]. One transporter gene (*chlG*) is located within the cluster. However, the author suggests that ChlG may not be efficient to transport the synthesized product out of the cells. Lobophorins are medium-sized spirotetronate antibiotics, and show antimicrobial and antitumor activity. The biosynthetic gene cluster for lobophorin was cloned from the deep-sea derived *Streptomyces* sp. [437]. The genes related to a transporter (*lobT1*) and the resistance (*lobT2*) are present within the cluster.

Platensimycin and platensin are novel class of antibiotics isolated from older microbial screening library [438,439]. These compounds show strong, Gram-positive antibacterial activity by selectively inhibiting cellular lipid biosynthesis [440]. The former inhibits β-ketoacyl carrier protein synthases (KAS) I/II (FabB/F), whereas the latter inhibits acyl carrier protein II/III (FabF/H). The biosynthetic gene clusters for platensimycin (PTM) and platencin (PTN) were cloned from *S. platensis* MA 7327, and *S. platensis* MA7339, respectively [441]. Four genes (*ptmP1*/ptnP1, *ptmP2*/*ptnP2*, *ptmP3*/*ptnP3*, and *ptnP4*/*ptnP4*) were proposed to be involved in the self-resistance. *ptmP4*/*ptnP4* codes for an efflux pump. PtmP3 is homologous to the FabB/F enzymes and can functionally replace both FabF and FabH enzymes of the housekeeping type II bacterial fatty acid synthase (FAS II) in *Streptomyces* species. In addition, FabF is resistant to platensimycin. Kalimantacins, linear polyketides, were isolated from *Alcaligenes* sp. YL-02632S and show strong antimicrobial activity against *S. aureus* and *S. epidemidis* including multi-resistant strains by using FabI, enoyl-acyl carrier protein reductase, as the target [442]. The kalimantacin biosynthetic gene cluster was cloned from *Pseudomonas fluorescens* [443]. The kalimantacin-producer strain possesses an isoform of FabI, BatG, within the biosynthetic gene cluster which confers full resistance to the producer and functionally complements the *E. coli fabI* mutation [444]. In addition, the author proposed the involvement of BatM, an alcohol dehydrogenase, in the self-resistance, where inactive 17-hydroxyl kalimantacin is exported out of the cells and then the inactive intermediate is activated to kalimantacin by the secreted BatM.

The rifamycins belong to the ansamycin group antibiotics produced by *Amycolatopsis mediterranei*. Rifampicin, a semisynthetic derivative of rifamycin, has been used as an anti-tuberculosis drug for over several decades. These compounds inhibit DNA-dependent RNA synthesis by binding to the DNA-dependent RNA polymerases of prokaryotes. The rifamycin biosynthetic gene cluster was cloned from *Amycolatopsis mediterrane* [445]. Three transporter genes (*orf21*, *orf22* and *orf23*) are located at one end of the cluster, whereas *rifP* encoding an efflux transporter is present in the middle of the cluster. However, the predominant cause for the self-resistance is due to the intrinsically resistant RNA polymerase encoded by *rpoB* located at the other end of the cluster. Three amino acid residues in RpoB were reported to be sufficient to confer rifampicin resistance to the producer (GB No.AF040570). The clinical isolates of rifamycin-resistant *M. tuberculosis* and *S. aureus* also possess mutant RNA polymerases, although multidrug transporters function as the resistance mechanism in minor parts of clinical resistant isolates [446]. Rifamycin phosphotransferase is also involved in the resistance in clinical isolates [447]. Holomycin is a member of the dithiopyrrolone class antibiotic produced by several *Streptomyces* species as well as Gram-negative bacteria such as *Yersinia ruckeri*. It possesses a broad spectrum of antimicrobial activity against Gram-positive and Gram-negative bacteria and inhibits RNA polymerases [448]. The holomycin biosynthetic gene clusters were cloned from *S. clavuligerus*, *Yersinia ruckeri* and marine bacterium *Pseudoalteromonas* sp. [449,450,451]. In *S. clavuligerus*, two MFS transporters (SSCG_03491/*hlmH* and SSCG_03542), flavin-dependent dihydroholomycin oxidase/disulfide reductase (SSCG_03492/*hlmI*), and S-methylation were proposed to be involved in the self-resistance [449,452]. In contrast, rRNA methyltransferase gene (*hom12*) is present within the gene cluster of *Y. ruckeri* besides MFS transporter gene (*hom8*) [450].

Salinosporamide A is a natural proteasome inhibitor isolated from the marine actinobacterium *Salinispora tropica* and is a promising clinical agent for the treatment of multiple myeloma. The salinosporamide A biosynthetic gene cluster was cloned from *S*. *tropica* CNB-440 [453]. The 20S proteasome machinery of *S. tropica* possesses a redundant proteasome β-subunit (SalI) within the salinosporamide A biosynthetic gene cluster, which confers 30-fold resistance to salinosporamide A in the producer species [454]. Up-regulation of proteasome subunits and the mutations of the β5-subunit encoding gene *PSMB5* have been observed in tumor cell lines with acquired resistance. Edeines are a group of closely related linear peptides produced by the soil bacterium *Brevibacillus brevis* and show an antibacterial activity against Gram-positive and Gram-negative bacteria. The mode of action of edeines was reported to inhibit DNA synthesis at low concentration, and translation at high concentration, and also inhibit cell division [455]. The edeine biosynthetic gene cluster was cloned from *B*. *brevis* Vm4 [456]. The transporter gene (*edeA*) is present at one end, whereas N-acetyltransferase gene (*edeQ*) is located at another end of the cluster, which acetylates the α-amino group of the 2,3-diaminopropionic acid residue of edeines. Zwittermicin A is the linear aminopolyol antibiotic produced by *Bacillus* species having the ability to suppress plant diseases and moderate activity against Gram-positive and Gram-negative bacteria. The biosynthetic gene cluster for zwittermicin A was cloned from *B. cereus* and *B. thuringiensis* [457,458]. *zmaR* encoding the N-acetyltransferase was reported to be involved in the self-resistance in *B. cereus* and *B. thuringiensis*, and *zmaWXY* encoding three transporters are located at the end of the cluster of *B. thuringiensis*.

Griseofulvin has been used for many years for the oral treatment of *Tinea capitis* and other dermatophyte infections [459]. Recent studies have paid attention to its antiviral and anticancer effects. The compound inhibits mitosis by affecting mitotic spindle microtubule functions. The griseofulvin biosynthetic gene cluster was cloned from *Penicillium aethiopicum* [460]. The cluster contains the chlorinase gene (*gsfI*) involved in the biosynthesis of griseofulvin, and the transporter gene (*gsfJ*). Mycophenolic acid is a fungal metabolite isolated as early as 1893. It was reported to possess antiviral, antifungal, antibacterial, antitumor, and immunosuppressive activity [461]. The target of mycophenolic acid is inosine-5′-monoposphate (IMP) dehydrogenase, which catalyzes the rate limiting step in the guanine nucleotide biosynthesis. The biosynthetic gene cluster for mycophenolic acid was cloned from *Penicillium brevicompactum* [462]. The gene cluster contains extra copy of IMP dehydrogenase gene (*mpaF*), which is resistant to mycophenolic acid. Interestingly, subgenus *Penicillium* species of both mycophenolic acid producers and non-producers possess two copies of IMP dehydrogenases [463].

## 6. Conclusions

The environment surrounding multidrug resistant pathogenic bacteria is getting worse and worse. The antibiotic resistance genes are complex mixtures of the genes of intrinsic antibiotic resistance [464,465], acquired resistance [466], and adaptive resistance [4]. These genes constitute the resistome [467,468,469]. This paper compares the resistance mechanisms in antibiotic producers and those in pathogenic bacteria. As a result, some points should be emphasized as follows. First, whereas some of the amino acid sequences of aminoglycoside acetyltransferases and phosphotransferases in pathogenic bacteria show high similarity E values, some others show no similarity at all (Appendix A). The similar phenomena are observed at their nucleotide sequence levels. The aminoglycoside acetyltransferase Eis, which is expressed in extensively drug-resistant strains of *M. tuberculosis*, is an exception, although they are present in other *Actinobacteria* as well as *Anabaena variabilis*, *B. anthracis*, and *E. faecalis* [81,82,470]. However, no report has been published on the involvement of these enzymes in self-resistance of the producers. As for β-lactamases, sequence similarities are observed between those from Gram-negative bacteria and *Actinobacteria* at amino acid and nucleotide levels (Appendix A) [5,209,234].

Second, some resistance mechanisms are observed in pathogenic bacteria, but not in antibiotic-producers. For example, the following mechanisms are recognized in pathogenic bacteria: mutations of nucleotides in 23S rRNA and ribosomal proteins in macrolide antibiotics [471,472,473], and macrolide esterases [152,153]; xanthine-guanine phosphoribosyltransferases and flavin-dependent mono-oxygenases in tetracyclines [161,474,475]; and nisin resistance protein/membrane-associated protease, modification of cell wall/membrane components in lantibiotics [310,311,312,313,314].

Third, according to the antibiotic classes, the bacteria exert their characteristic resistance mechanisms; antibiotic modifications such as acetylation and phosphorylation in aminoglycoside antibiotics; antibiotic modification by glycosylation and target-modification by rRNA methylation in macrolide antibiotics; antibiotic destruction by β-lactamases and target-modification of penicillin-binding proteins in β-lactam antibiotics; sequestration by binding to specific proteins in DNA interacting antitumor antibiotics such as bleomycin group of antibiotics [338,340,341], mitomycin [375], thiocoraline [393], and enediyne antibiotics [476]; DNA repair in DNA interacting antitumor antibiotics such as nogalamycin, yatakemycin, chromomycin A_3_, mithramycin, echinomycin, and enediyne antibiotics.

Fourth, some antibiotic producing bacteria possess two target enzymes, and at least one of them is resistant to their own antibiotics. For example, the indolmycin-producer strain possesses two tryptophanyl-tRNA synthetase genes, and one of them is resistant to indolmycin [386]; the novobiocin-producer strain possesses topoisomerase (DNA gyrase)-sensitive and resistant genes, *gyrB^S^* and *gyrB^R^* [405]; courmermycin A_1_-producer strain has two coumermycin- and clorobiocin-resistant topoisomerase genes, *gyrB^R^* and *parY^R^* [408]; salinosporamide A-producer strain holds a salinosporamide-resistant redundant proteasome β-subunit *salI* gene within the biosynthetic gene cluster [454]; and the mycophenolic acid-producer strain possesses mycophenolic acid-resistant extra copy of IMP dehydrogenase gene *mpaF* within the biosynthetic gene cluster [462].

Fifth, in accordance with the dissemination of multidrug resistant pathogenic bacteria, the acquisition of multidrug resistant determinants in opportunistic and/or resident pathogens, and the discovery and introduction of new types of antibiotics such as enediyne compounds, new types of resistance mechanisms such as the self-sacrifice proteins [424,427,477], aminoglycoside acetyltransferases ’Eis’ [81,82], and adaptive antibiotic resistance [4] have emerged, and in addition the transfer of the resistance mechanism in producers to new fields of pathogenic bacteria such as 16S rRNA methyltransferases were reported [95,96]. I am worrying that these types of mechanisms will be prevailing in the whole environments near future.

Antibiotic resistance can develop through three distinct mechanisms: intrinsic resistance, acquired resistance and adaptive resistance [2,3,4]. Adaptive resistance can emerge as a result of concentration gradient and contact with sub-inhibitory concentrations of antibiotics [4]. Among three mechanisms, antibiotic resistances acquired by horizontal gene transfer poses a major threat to human and livestock [478,479]. To establish the successful transfer from donor bacteria to recipient bacteria, that reside in various environments, by transformation, transduction or conjugation [7,8,9], it is necessary to overcome the barrier of GC-content between bacterial species [480,481,482,483,484]. These authors described that a large excess of synonymous GC→AT mutation over AT→GC mutations arises across a broad range of phylogenetically diverse species, that in a wide variety of bacterial species, the evolution of GC-content by recombination tends to increase the probability of fixation of AT→GC mutations, and homologous recombination via GC-biased gene conversion (gBGC) is a crucial factor universally influencing the nucleotide content of genes and genomes, and that genes capable of conferring antibiotic resistance are not easily transferred to human pathogens, especially if selection is absent. That is that the GC-content of the genome is a result of complex shuffling of various DNA fragments by mutation, recombination and selection. As a result of these phenomena, GC-contents in the genomes distribute broardly from 16.5% of *Carsonella ruddii* to 74.9% of *Anaeromyxobacter dehalogenans* in bacterial species [485,486]. However, recent analyses of gene transfer to *E. coli* using 200 genes in human pathogens varying widely in resistance mechanisms, targeted antibiotic classes, and phylogenetic dissemination revealed that sequence composition such as codon adaptation index (CAI), GC-content, N-terminal mRNA-folding energy, and gene length is not a major functional barrier, but resistance mechanism and phylogenetic origin are more important determinants deciding the functional compatibility and fitness of the antibiotic resistance genes in the recipient bacteria [147]. Furthermore, phylogenetic relatedness of the donor and the recipient species affects the functional compatibility and fitness cost of newly acquired genes. These results are supported by the analyses of the antibiotic resistance gene distribution in 17 important human pathogens [484]. Considering these results, it is possible to conclude that the environments surrounding the antibiotic resistance in human and livestock have changed from the early antibiotic phase to the newer antibiotic phase. At the early antibiotic phase, even though the transfer of antibiotic resistance genes happened among the antibiotic producing soil bacteria, non-producing environmental bacteria and the clinic pathogens, the frequency and the technique including mobile genetic elements are more restricted and regulated. Moreover, it sometimes is possible to predict the future environment encompassing the antibiotic resistance. However, as the usage and the classes of antibiotics have been enlarged and entered into a new generation, the resistome is getting more and more complex and sophisticated and, is also entering into a new generation [468,469]. In addition, it is going harder and harder to predict the future of antibiotic resistance [487,488]. So when we use antibiotics, we should follow the recommendation of WHO at least [1].

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
