# Peer review of "Comparison of Antibiotic Resistance Mechanisms in Antibiotic-Producing and Pathogenic Bacteria"

_molecules, 2019, doi:10.3390/molecules24193430_

Round 1

Reviewer 1 Report

Review of the article: „Comparison of Antibiotic Resistance Mechanisms in Producing and Pathogenic Bacteria”

Manuscript ID - molecules-584086

The amount of material (publications, nucleotide and amino acid sequences) analysed by the author of the manuscript is impressive. The choice of topic is very interesting - the transfer of genes that determine antibiotic resistance is extremely important, primarily from a clinical point of view. In my opinion, the article can be accepted, only minor corrections are required. Although, in some subsections, the author mainly presented the mechanisms of antibiotic activity and resistance that exist in producing strains - the resistance mechanisms of producing strains and pathogenic bacteria were not really compared (or this comparison was not clearly presented).

Detailed comments:

Title – no comment, the proposed title is acceptable

Abstract – well written. The author briefly, presented the conception of the text of the article. Four most important points are discussed. However, I have some doubts about point two (lines 14-17): “Second, the antibiotic resistance genes in pathogens are compared with those in the producers. Resistance mechanisms have dependency on antibiotic classes. New types of resistance mechanisms such as Eis aminoglycoside acetyltransferase and self-sacrifice proteins in enediyne antibiotics emerged in pathogens”. In fact these are three separate sentences. In my opinion the author should modify this part – to combine these three sentences. The abbreviation GC should be explained

Introduction – I have only one question to the author – would it be possible to present any statistic data that support the information presented in the firs sentence of introduction: “The introduction of antibiotics once reduced human morbidity and mortality caused by infectious diseases dramatically”. Of course it is not obligatory, it is only my suggestion (e.g. presenting differences in infant mortality today and in the pre-antibiotic era, or in countries where antibiotics are available and rationally used, and in countries where access to medicine is very limited could show the readers that the problem is really important)

Main text of the manuscript

Aminoglycosides

One general comment – figures 1-4 (phylogenetic trees are well prepared), but there is a problem with tables. They are large, contain characteristic of many different antibiotics (in my opinion they are absolutely necessary in the text of the manuscript), but technically they are not well prepared, for me it is difficult to read information in these tables (particularly Table 2 should be changed).

The classes of antibiotics (lines57-58) should be pointed (e.g. 1) …, 2) …)

Line 153 Actinobacteria – should be written with ithalic

The analysis of mechanisms of resistance against aminoglycosides (pages 2-16, including two tables and 4 dendrograms) is particularly impressive and obtained results are very interesting. The presented by the author cluster analysis of four targets have shown some important differences between nucleotide/amino acid sequences of genes/enzymes responsible for resistance against these antibiotics in producing and pathogenic bacteria. It is a very important observation, which in fact is inconsistent with the opinions (that are still “valid” and very common) that pathogenic bacteria acquire resistance mechanisms from producing bacteria.

Macrolide antibiotics

Interesting are also conclusions coming from investigation of mechanisms of resistance against macrolide antibiotics: “rRNA methyltransferases and phosphotransferases in pathogenic bacteria are closely related to those in macrolide-producing bacteria, whereas rRNA mutations and efflux pumps are scarcely related with each other. These resistance characters may have been transferred from other sources.” Moreover, pathogenic bacteria produce also macrolide esterases, which are not used for self-protection of producing microorganisms.

Tetracycline and chloramphenicol – no critical comments nor suggestions

Other protein synthesis inhibitors – in my opinion a general conclusion should be presented at the end of this fragment of the manuscript (if there is an important link between the resistance of producing strains and pathogenic bacteria)

Beta-lactams and Glycopeptides, lipopeptides and related antibiotics – no critical comments

Polyene macrolides – from my best knowledge resistance to Amphotericin B is extremely rarely observed and it is not a consequence of overproduction of drug efflux transporters. A clearer conclusion should be presented at the end of this subchapter. However I am not sure if situation is not a bit different in comparison to other discussed antibiotics. 

Lantibiotics and cyclic peptides – well presented, and conclusion “Summarizing these results, the mechanisms of action of lantibiotics in pathogenic bacteria are little different from those in producing bacteria. In producing bacteria the main players are ABC transporters and immunity protein, while in pathogenic bacteria the nisin resistance protein and the modification of cell wall are additional players.” is clear and informative for readers. The similar summary I would expect in some other fragments – please look my comment above.

Other cell wall/membrane synthesis inhibitors – the author presented most important antibiotics from this group, mechanisms of their activity and mechanisms of resistance that exist in producing strains. But there is not clear information if these mechanisms were transferred to pathogenic microorganisms (e.g. M. tuberculosis). I would be grateful for a short comment.

Bleomycin and related antitumor antibiotics – I am not really sure if antitumor antibiotics should be included to the paper – it depends of author and editor decision. The same comment about next two subchapters - Quinone and related antitumor and antimicrobial antibiotics and Enediyne antitumor antibiotics.

In the section: “Other antibiotics” definitely the best description is presented for rifampicin. In this description a link between mechanisms of resistance of producing strains and pathogenic bacteria (S. aureus and M. tuberculosis) is clearly indicated.

Conclusions – well presented, no comments   

Final decision – the manuscript can be accepted after minor revision

Author Response

Dear Sir/Madame,

Thank you very much for the valuable comments to the manuscript ID Molecule-584086. According to the comments, the following answers and corrections were made.

Reviewer #1.

In some subsection, the resistance mechanisms were not clearly compared in producing strains and pathogenic bacteria. This is mainly because the resistance mechanism in pathogenic bacteria has not been definitely clarified.
As for Abstract. Three sentences were combined as follows: “Second, when the antibiotic resistance genes in pathogens are compared with those in the producers, resistance mechanisms have dependency on antibiotic classes, and, in addition, new types of resistance mechanisms such as Eis aminoglycoside acetyltransferase and self-sacrifice proteins in enediyne antibiotics emerged in pathogens.”
As for Abstract. GC was changed to “guanine + cytosine (GC)”.
As for Introduction. The sentence “For example, human morbidity and mortality by tuberculosis were greatly reduced after the introduction of streptomycin and kanamycin.” was inserted in Introduction.
As for Aminoglycosides. All the figures and tables were transferred to the supplementary files. Table 2 shows the amino acid sequence similarities between the amino acid sequences in the vertical axis and those in the horizontal axis. In addition, the sentence “Antibiotic producers are marked with red.” was inserted in the legend. I don’t know how to change these data.
As for Aminoglycosides. The classes of antibiotics were written in bold letters. In addition, empty rows were inserted between different classes in Table S1 (formerly Table 1).
As for Aminoglycosides. “Actinobacteria” was italicized: Lines 76, 140, 198, 271, 299, 464, 471, 609, 797, 810, 819, 1384 and 1387,
As for Other protein synthesis inhibitors. The sentence “The resistance mechanisms in this class of antibiotics are generally similar between producers and pathogens, although the details have not been elucidated yet.” was inserted at the end of subsection.
As for Polyene macrolides. As described in the text, the resistance to amphotericin B in clinical isolates of Aspergillus terreus is mainly due to the membrane modification such as ergosterol or lower ergosterol content. However, other mechanisms are thought to be implicated as described in the last paragraph of the subsection.
As for Lantibiotics and cyclic peptides. Unfortunately, however, in other antibiotic classes no such clear difference has been detected between producers and pathogens.
As for Other cell wall/membrane synthesis inhibitors. The sentence “In M. tuberculosis, mutations of alanine racemase, L-alanine dehydrogenase are reported to be implicated in the resistance [332,333]. On the other hand, fosfomycin resistance is very rare in pathogenic bacteria [334].” was added at the end of this subsection. Three new references were added. The transferability of the resistance genes has not been described yet.
As for Bleomycin and related antitumor antibiotics. Most of these antibiotics show antimicrobial activity as well as antitumor activity. The reasons that I included these subsections to this review paper are following two main points. One is that the resistance mechanisms in these producers are clearly different for those in other antibiotics. Second, I am worrying about the dissemination of these types of resistance mechanisms to pathogens in addition to tumor cells near future

Reviewer 2 Report

In the manuscript „Comparison of Antibiotic Resistance Mechanisms in Producing and Pathogenic Bacteria“ of Hiroshi Ogawara, the author reported on characteristics of different mechanisms involved in antibiotic resistance mechanisms. Therefore, Ogaware disussed different issues: (i) antibiotic resistance genes in producersis related to its biosynthesis, (ii) antibiotic resistance genes in pathogens vs. producers, (iii) the relationships of the resistance genes between producers and pathogen as well as (iv) the GC barrier in gene transfer to pathogenic bacteria.

It is the opinion of this reviewer that the provided manuscript of Ogawara povide a carefully summarized overview with a detailed discussion on the reported issues. I recommed that the figures and the tabels should be removed from the manuscript and included in a supplemental material section. This will provide the information of the whole text in a more readable form for interested readers. The manuscript will benefit from this because currently it is very long.

Verall, all information are carefully summarized and the manuscript is well written.

Minor comments:

Please evaluate the use of antibiotic resistance vs. antimicrobial resistance in the whole manuscript!

Author Response

Dear Sir/Madame,

Thank you very much for the valuable comments to the manuscript ID Molecule-584086. According to the comments, the following answers and corrections were made.

Reviewer #2.

All the figures and tables were transferred from the text to the supplementary files.
I use “antibiotic resistance” and “antimicrobial resistance” as almost the same meaning.

Reviewer 3 Report

The topic is very interesting and I appreciated the amount of details, notwithstanding  I have some major concerns listed below:

the manuscript lists the gene clusters of the producers so far sequenced and the presence of antibiotic resistance genes and compare the resistance genes and not the resistance mechanisms with those of the pathogenic bacteria. On the basis of the title, I expected he evaluated the known mechanisms between producers and pathogens. While, in many cases, the mechanism is not known or no resistant pathogens have been isolated so far or antibiotics are not in use because of their low activity or other reasons. The same author wrote lines 50-52: “This review paper summarizes at first the antibiotic resistance genes in producer bacteria from the point of view of the antibiotic biosynthesis. Then the resistance genes in pathogenic bacteria are compared with those in the producers. Lastly, the relationships of the resistance genes between producer bacteria and  pathogenic bacteria are reevaluated again at their amino acid sequence as well as nucleotide sequence levels”. the manuscript is confusing and not well organized; the use of sub-titles or sub-paragraphs could be helpful to improve the reading. I would suggest to differently organize the information. For example, the author could dedicate a paragraph to the resistance mechanism to an antibiotic and inside it to discuss on the comparison between producers and pathogens. In the present version, he lists all the genes and the proteins of the producers and then those of the pathogens, with some comparison at gene and protein level and the focus on the mechanism is lost.

 My minor comments/suggestions:

Title:

I think that if you want to describe mechanisms you should eliminate all the parts on antibiotics for which the mechanism is not established yet and the parts concerning the antibiotic for which no resistant strains have been isolated so far. Otherwise, rephrase the title.

In addition, you should mention antibiotic producers and not only producers.

In the abstract, you wrote “This paper deals with this problem from four points.” Not clear the fourth. And you wrote: “between early antibiotic phase” not clear, you could write “early stage of antibiotic use” and please write the whole name of the genus of bacteria the first time, ie S. aureus.

L 49, 51 Please use producer as a noun and producing as adjective, not producer bacteria.

L41-43 the enzymes are not transferred and please check the verb.

Table 1: please add the class of the antibiotics, maybe an empty row is sufficient

L183 was assumed to play

L190 genes and not proteins are present in a cluster

L191 check acarobose

L199-201 These two sentences are not very nice. Please rephrase them.

L214 enzymes…. were not was

L226-227 change in “reflecting the different guanine+cytosine content of their genome.”

In the figures could you differentiate the producers from the pathogens with a different color or underlining the text?

L584 “based on the chemicalstructuresof the number of atoms” is right?

L594 It contains genes for..

L692  as the amino acid sequences are highly..

L712 the mutations in these nucleotides have been found in many ..

L722 streptogramin

L741 This type of resistance has not been

L827 check N.cardia lactamdurans

829 check kirromycin

L836-837 sentence not clear

L846 as fas as I know GE2270 is produced by Planobispora rosea

L862-863 not clear 5 or 6?

L889  check a various carbapenem compounds

L922 genes not enzymes are detectable in its genome

L932 please use the entire words instead of MICs

L938 use the singular

L996 and L1261 teicoplanin not teichoplanin

L1003 from reaching the target that is not inside the cells, thus the verb enter is misleading

L1019 please check carefully the bibliography, as far as I know bal cluster does contain vanSR genes

L1031-1033 not clear

L1044-1046 this information was already given, L1001-1003

L1157 Microbispora corallina

L1191 rephrase Thirdisthe modifications of cell wall/membrane.

Paragraph 3.5 Resistant strains to capuramycin have been isolated?

L1294 The doxorubicin (14-hydroxydaunorubicin, adriamycin) and daunorubicin biosynthetic gene clusters were cloned

L1311 is the question mark necessary?

Conclusion should be rewritten on the basis of my previous comments.

Author Response

Dear Sir/Madame,                                                

Thank you very much for the valuable comments to the manuscript ID Molecule-584086. According to the comments, the following answers and corrections were made.

Reviewer #3.

The manuscript does not discuss the mechanisms of antibiotics, but compares the resistance mechanisms of antibiotic producers and pathogenic bacteria. Before this, the manuscript lists the strategies of the resistance mechanisms to their own antibiotics in antibiotic producers. The self-resistance genes are usually located within or adjacent to their biosynthetic gene clusters as shown in Table S1. As two other reviewers described in their comments that the manuscript is well organized and is well written, I should like to follow their comments. In addition, I believe that the manuscript is well organized and well written, not confusing. As for Title. “Antibiotic” was inserted. As for Abstract. Forth is “Lastly”. “early antibiotic phase” was changed to “early stage of antibiotic use”. “S. aureus” was changed to “Staphylococcus aureus”. “producer” was changed to “producing”. Lines 51, 53, 452 and 458, Line 44. “genes for ” was inserted. Table S1 (formerly Table 1). Empty rows were inserted between different classes of antibiotics. Line 111. “genes for” was inserted. Line 171. “plays” was changed to “play”. Line 179. “Acarobose” was changed to “Acarbose”. Line 188. “affects not only” was changed to “not only affects”. Line 202. “was” was changed to “were”. Line 214. “different” was inserted. In the figures, the producers and pathogens were differentiated by color. Line 356. “the number of atoms” is right. Line 366. “They contain” was changed to “It contains”. Line 464. “is” was changed to “are”. Line 484. “mutation” was changed to “mutations”. Line 494. “stretptogramin” was changed to “streptogramin”. Line 513. “have” was changed to “has”. Line 599. “N.cardia” was changed to “N”. Lines 601 and 605. “kirromysin” was changed to “kirromycin”. Line 608. “was” was changed to “is”. Line 319-320. GE2270 is produced by Planobispora rosea, but is also produced by Nonomuraea sp. WU8817 and the gene cluster was cloned from the latter species (GenBank No. FJ461359). Line 637. 5 not 6. Nocardicin A and sulfazecin are monobactam antibiotics. Line 664. Imipenem and meropenem are carbapenem compounds. Line 697. “genes for” was inserted. Line 707. “MICs” was changed to “minimum inhibitory concentrations (MICs)”. Line 713. Why the singular? I think “ampicillin and cephem group β-lactams” is a right phrase. Lines 772 and 1039. “teichoplanin” was changed to “teicoplanin”. Line 779. “entering the cells” was changed to “reaching the target”. Line 795. vanSR genes are present in balhimycin producer but are located about 2Mb apart from vanHAX cluster, but not in the DNA flanking the vanHAX cluster. Line 808. “which is” was inserted. Line 822. “as described above” was inserted. And reference [257] is not shown in the above sentence. Line 933. “coralline” was changed to “corallina”. Line 967. “cell wall/membrane” was changed to “cell wall and membrane”. Line 1004. Only side chain (H or methyl) is different between capuramycin and A500359A. Line 1072. “was” was changed to “were”. Line 1089. SnoO has not definitely identified as a polyketide cyclase. As for Conclusion. As described in answers to comments 1, I don’t think I need to rewrite the conclusion.

Round 2

Reviewer 2 Report

The manuscript is now acceptable for publication.

Author Response

Thank you for your kind review.

Reviewer 3 Report

I still think that the title does not exactly correspond to the content of the manuscript.

Many other comments and suggestions were accepted by the author, even if the manuscript has not been changed from the last version.

Author Response

Thank you for your comments.